# PANINI: Continual Learning in Token Space via Structured Memory

**Shreyas Rajesh** [* 1]  **Pavan Holur** [* 1]  **Mehmet Yigit Turali** [1]  **Chenda Duan** [1]  **Vwani Roychowdhury** [1]

## Abstract

Language models are increasingly used to reason over content they were not trained on, such as new documents, evolving knowledge, and user-specific data. A common approach is retrieval-augmented generation (RAG), which stores verbatim documents externally (as chunks) and retrieves only a relevant subset at inference time for an LLM to reason over. However, this results in inefficient usage of test-time compute (LLM repeatedly reasons over the same documents); moreover, chunk retrieval can inject irrelevant context that increases unsupported generation. We propose a *non-parametric continual learning* framework, where the base model remains fixed, and learning occurs by integrating each new experience into an external semantic memory state that accumulates and consolidates itself continually. We present PANINI, which realizes this by representing documents as Generative Semantic Workspaces (GSW)—an entity- and event-aware network of question–answer (QA) pairs, sufficient for an LLM to reconstruct the experienced situations and mine latent knowledge via reasoning-grounded inference chains on the network. Given a query, PANINI *only* traverses the continually-updated GSW (*not the verbatim documents or chunks*), and retrieves the most likely inference chains. Across six QA benchmarks, PANINI achieves the highest average performance, 5%–7% higher than other competitive baselines, while using 2–30× *fewer* answer-context tokens, supports fully open-source pipelines, and reduces unsupported answers on curated unanswerable queries. The results show that efficient and accurate structuring of experiences at write time — as achieved by the GSW framework — yields

both efficiency and reliability gains at read time. Code is available at https://github.com/roychowdhuryresearch/gsw-memory.

## 1. Introduction

Language models have become powerful general-purpose reasoners (Wei et al., 2023; Guo et al., 2025), and through their ability to adapt to new tasks and information *in-context* (Kojima et al., 2023), are increasingly used to reason over content they were not trained on—such as new documents, evolving knowledge, and user-specific data. While in-context learning is effective, it faces fundamental scalability limits: as the amount of information grows, maintaining strong downstream performance becomes increasingly difficult, since context windows remain bounded and long-context failure modes such as lost-in-the-middle (Liu et al., 2024) and context rot (Hong et al., 2025) emerge with scaling. *Parametric continual learning (PCL)*, is a natural option (Chen et al., 2019): as significant patterns in new experiences emerge, instead of repeatedly re-injecting the same information into the prompt, one can encode such new patterns into the model itself, i.e. by updating model parameters on relevant data via a suitably-defined loss function. While such methods have a long history (McCloskey & Cohen, 1989) and recent test-time training/adaptation approaches for LLMs show increasingly strong capabilities (Hu et al., 2025; Yuksekgonul et al., 2026), they introduce substantial challenges: repeated (often expensive) training runs and data curation, risks of catastrophic forgetting (Huang et al., 2024), and poor interactions with the multi-stage post-training (instruction tuning, preference alignment) that modern LLMs undergo (Qi et al., 2023; Lin et al., 2024). More broadly, *there is not yet a generally reliable and effective way to disentangle post-training effects*, continually train, and then restore aligned instruction-following behavior without re-running post-training or requiring paired checkpoints (Huang et al., 2025; Djuhera et al., 2025).

These constraints motivate *non-parametric continual learning (NPCL)*: methods that keep the base model fixed and store new documents and experiences externally, retrieving relevant evidence at inference time. *What would a genuine NPCL system look like?* Consider how humans continually adapt with new information: we integrate each new

---

[*]Equal contribution [1]Department of Electrical and Computer Engineering, University of California, Los Angeles, USA. Correspondence to: Shreyas Rajesh <shreyasrajesh38@ucla.edu>, Vwani Roychowdhury <vwani@ucla.edu>.

*Proceedings of the 43rd International Conference on Machine Learning*, Seoul, South Korea. PMLR 306, 2026. Copyright 2026 by the author(s).

experience into memory—consolidating, linking to prior knowledge, and forming structured representations at *encoding time* that reduce future effort and improve reliability during *retrieval time*.

Thus, an NPCL framework must make two inter-related design choices: **(a)** *Cumulative memory architecture*: what gets written into memory, especially as the number of documents/experiences scales and **(b)** *Reading Memory at QA:* Query-specific retrieval of memory in **(a)** to minimize test-time compute and maximize accuracy. We argue the quality of these choices should be measured against three criteria: **(i)** the ability to synthesize and reason over stored experience to answer *supported* questions **(ii)** *efficiency of retrieval* as the experience store grows; and **(iii)** ability to recognize when stored experience *does not support* an answer and abstain. These criteria distinguish a continual learner from a system that merely retrieves and answers queries.

The most common practice in implementing NPCL is retrieval-augmented generation (RAG)(Lewis et al., 2021; Karpukhin et al., 2020), which stores documents as verbatim fragments and retrieves relevant passages at query time. Within this space, work has targeted improving both design choices, though the vast majority has focused on **(b)**– improving retrieval. This includes improved dense encoders (Lee et al., 2025; Zhang et al., 2025), agentic methods that interleave reasoning and retrieval (Jin et al., 2025; Trivedi et al., 2023) and even graph-based methods like HippoRAG (Gutiérrez et al., 2025a;b) *that build a knowledge graph* for efficient traversal but still feed verbatim passages to the LLM. While sophisticated, these RAG systems force LLMs to re-process the same chunks repeatedly, paying the full inference cost each time. Moreover, as our results show (Tables 2, 3) chunk-based retrieval can inject irrelevant context that increases unsupported and hallucinatory generation.

A smaller set of methods do explicitly invest in **(a)**—transforming verbatim text into structured memory. RAPTOR (Sarthi et al., 2024) builds hierarchical summaries via recursive clustering; GraphRAG (Edge et al., 2025) extracts entities and relationships, clusters them into communities, and generates community-level summaries. While these approaches move beyond verbatim storage, their representations are optimized for compression and thematic summarization—answering questions like "what are the main themes?"— rather than for reasoning across linkages to retrieve specific latent knowledge.

To address the above challenges, we present PANINI, an NPCL framework (Fig. 1) that strikes a balance by investing in both design choices **(a)**, and **(b)**. For **(a)**, *what gets written*, it builds on the Generative Semantic Workspaces (GSW) representation (introduced by Rajesh et al. (2025); Holur et al. (2024)): an entity- and event-aware network of question–answer pairs—structured memory sufficient for an

LLM to reconstruct experienced situations and mine latent knowledge via reasoning-grounded inference chains. For **(b)**, *how memory is read*, we introduce Reasoning Inference Chain Retrieval (RICR), a beam-search style retrieval procedure that decomposes queries and follows reasoning chains through the GSW. As supported by extensive evaluation across six QA benchmarks spanning single-hop and multi-hop reasoning, PANINI outperforms competitive baselines on all three previously defined NPCL criteria: *supported performance* and *inference-time efficiency* (Table 2), and *reliable abstention* (Table 3).

The main contributions of this paper can be summarized as: (1) **Evaluation criteria for NPCL:** As discussed, we propose and quantify via extensive QA benchmarking three criteria any NPCL system should satisfy—supported performance, inference-time efficiency, and reliable abstention (2) **PANINI:** A novel NPCL framework combining GSW-based structured memory with our chain-following retrieval procedure (RICR), achieving SOTA performance at 5–30× fewer inference-time tokens. (3) **Reliability evaluation:** We curate evaluation splits for multi-hop QA benchmarks separating answerable from unanswerable questions to test abstention under missing evidence, on which PANINI outperforms all baselines.

## 2. Panini

To keep this section self contained, we first summarize the GSW and how the representation is indexed for efficient retrieval. We then present RICR, a beam-search like procedure that uses a single LLM call for question decomposition and follows entity chains through the GSW to accumulate evidence. Finally, we describe the answer generation step that grounds responses in the retrieved QA pairs, and outline the evaluation protocol we use to measure reliability under missing evidence.

Relative to prior GSW work on episodic memory, PANINI instantiates the representation for fact-centric multi-hop QA akin to declarative memory. Prior work reconciled entities and states across continuous narratives; here, each source document is short and fact-focused, so we keep document-local workspaces and avoid exhaustive cross-document reconciliation at write time. RICR instead creates only the querw-relevant cross-document links at read time by chaining through retrieved QA answers.

### 2.1. Structured Memory: Representation and Indexing

**Representation.** Let $\mathcal{D} = \{d_i\}_{i=1}^N$ denote a corpus of documents. We construct a Generative Semantic Workspace (GSW) for each document $d_i$ following Rajesh et al. (2025). Each GSW $\mathcal{G}_i = (E_i, V_i, Q_i)$ consists of (i) entity nodes $E_i$ with associated roles and states, (ii) verb-phrase/event nodes

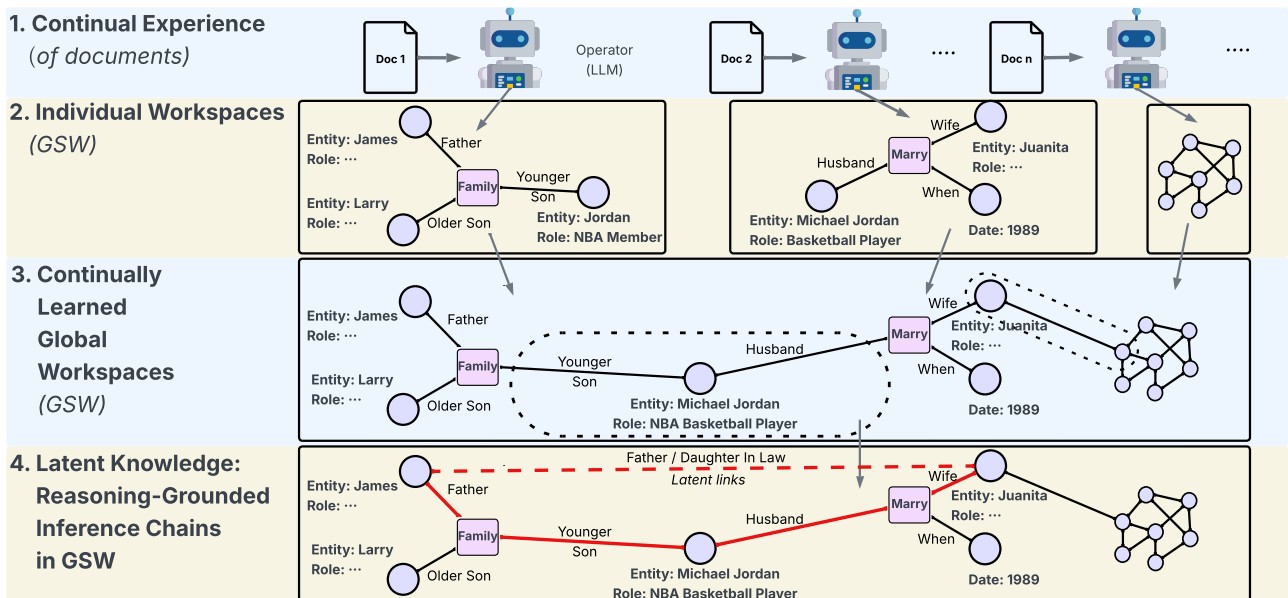

*Figure 1.* A non-parametric continual learning (NPCL) framework schematics (1) **Continual experience**: incoming documents are processed asynchronously, potentially by different agents. (2) **Individual workspaces**: each experience is encoded into a Generative Semantic Workspace (GSW). (3) **Continually learned global workspace**: GSWs can be continually consolidated by reconciling entities, events, and actions both across and within documents. Extensive ablation studies (see Table 10) show that different combinations of LLM models of different sizes for performing different tasks – GSW generation, and retrieval – lead to consistently robust performance. Thus GSW can be used as a shared meta-representation. (4) **Reasoning-grounded inference**: The goal is to have *enough reconciliation* –but not exhaustive– so that *all latent knowledge supported by the collection of experiences are represented by inference chains/paths.*

$V_i$ capturing actions and relations, and (iii) question–answer (QA) pairs $Q_i$ that *attach to verb phrases and point to entities*. Concretely, each QA pair is grounded in a verb-phrase node $v \in V_i$ and can be viewed as a directed labeled edge $(v \xrightarrow{q} e)$ where $q$ is a natural-language question specifying an attribute of the event and $e \in E_i$ is the entity node corresponding to the answer.

To illustrate, consider the statement: *"Barack Obama, the 44th President of the United States, was born on August 4, 1961 in Honolulu, Hawaii."* A triplet store would encode this via schema-bound facts such as (Obama, date_of_birth, 1961-08-04) and (Obama, place_of_birth, Honolulu). In contrast, a GSW represents this information as a small semantic network: an entity node for Barack Obama (role/state: 44th President), a verb-phrase node was born, and entity nodes for August 4, 1961 (state: birth date) and Honolulu, Hawaii. The QA pairs in $Q_i$ form the edges between the entities and the verb-phrases, e.g., *"When was Barack Obama born?"*→*"August 4, 1961"* and *"Where was Barack Obama born?"*→*"Honolulu, Hawaii"* (and inverse forms such as *"Who was born on August 4, 1961?"*→*"Barack Obama"*). This design makes the set of entities associated with an event explicit in short, atomic units, which we leverage in the next subsection for efficient retrieval. A full sample GSW can be found in Appendix B.

**Accessing GSW during retrieval: Dual Indexing.** We build two corpus-level indices over the per-document workspaces. First, we create a sparse BM25 index over *entities*, where each entry includes the entity surface form (Barack Obama) together with its associated role/state information from the GSW (44th President). Given a query, we score it against this entity index and map the top-ranked matches back to their corresponding entity nodes in the originating document-level GSWs; these matched entities serve as entry points for navigating the local semantic network and identifying the most relevant attached QA pairs. Second, we build a dense vector index over *all QA pairs* extracted from all GSWs, enabling semantic retrieval of candidate QA pairs for each question. Importantly, unlike chunk-based retrieval and many graph-based memories that return long passages or large neighborhoods, our inference-time evidence is composed *only of QA pairs* from the GSW, yielding compact, targeted factual support for downstream reasoning and question answering.

### 2.2. Reasoning inference chain retrieval (RICR)

This subsection describes how PANINI *reads* from the structured memory store to answer a query. Given an input query $q$, the system (i) produces a short plan consisting of *atomic* sub-questions, (ii) assembles a compact set of grounded QA pairs by following entity-linked connections in the mem-

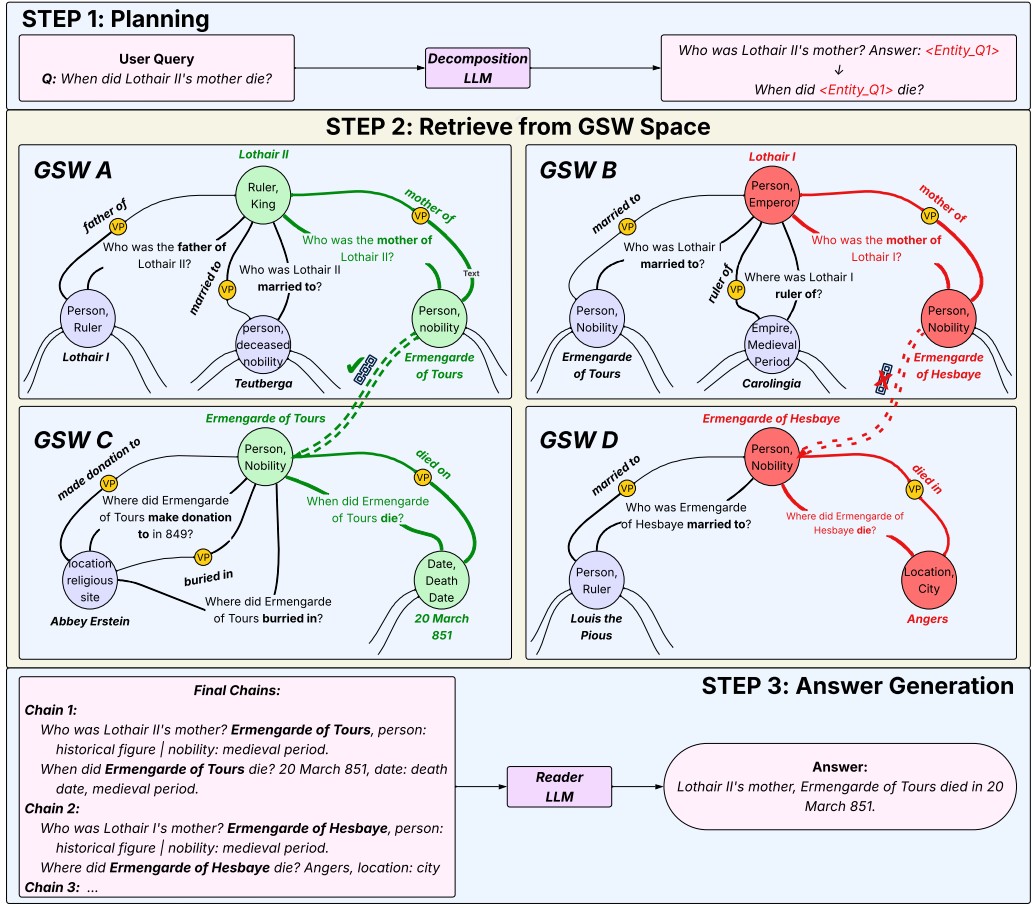

*Figure 2.* System overview of PANINI at inference time. **Step1: Planning:** A decomposition LLM converts the user query into an ordered sequence of single-hop sub-questions. **Step2: RICR:** We perform chain-based retrieval by expanding candidate paths hop-by-hop: starting from a seed entity (here, *Lothair II* and *Lothair I*), it follows QA edges to propose intermediate entities (e.g., candidate mothers) and incrementally extends partial chains across GSWs. Candidate chains are scored at each hop and low scoring paths are pruned. **Step3: Answer Generation:** The top-ranked chains are de-duplicated and provided to the final answering LLM.

ory, and (iii) generates a final answer conditioned on the retrieved QA evidence.

**Planning.** We instantiate a planning module DECOMPOSE($\cdot$) that rewrites an input query $q$ into one or more parallel sequences of atomic sub-questions $\{q_{i,t}(x_{i,t})\}$. Each sub-question $q_{i,t}(\cdot)$ takes an argument $x_{i,t}$: for initial sub-questions, $x_{i,1}$ is a static entity from the query; for subsequent sub-questions, $x_{i,t} = a_{i,t-1}$, the answer to the preceding sub-question in that sequence. Each sub-question targets a single fact that can be resolved by QA pairs from the GSW. The subscript $i$ in $q_{i,t}(\cdot)$ denotes different parallel QA decomposition question sequences often needed for more complex queries.

For example, *"Who died later, the mother of Lothair II or the father of Amadeus I?"* produces two parallel sequences: (i) $q_{1,1}$(Lothair II) = *"Who was the mother of Lothair II?"* → $q_{1,2}(a_{1,1})$ = *"When did $a_{1,1}$ die?"*; (ii) $q_{2,1}$(Amadeus I) = *"Who was the father of Amadeus I?"* → $q_{2,2}(a_{2,1})$ =

*"When did $a_{2,1}$ die?"*.

Unlike agentic systems that interleave decomposition and retrieval, PANINI decomposes *once*; all subsequent steps are non-parametric retrieval and scoring.

**RICR.** Given the decomposed sub-questions $\{q_{i,t}(x_{i,t})\}$, RICR retrieves evidence by iteratively executing retrieval *hops* and assembling the results into scored chains.

**Hops.** A *hop* is an atomic retrieval step that resolves one sub-question. Given an instantiated query $q_{i,t}(x_{i,t})$, the hop searches the GSW and retrieves a QA pair whose answer $a_{i,t}$ can instantiate subsequent sub-questions in the same sequence.

**Hop Subroutine: Retrieve and Score** (the RETRIEVEAND-SCORE subroutine used in Algorithm 1). Each hop executes the following retrieval subroutine. Given a sub-question $q_{i,t}(x_{i,t})$, we retrieve candidate GSW QA pairs $(q^G, a^G)$ using a dual search approach: (i) query the BM25 entity

index to obtain high-scoring entity nodes, then collect the QA pairs attached to those entities; and (ii) query the dense QA-pair index to retrieve semantically similar QA pairs directly. The candidates are merged and reranked using a cross-encoder against $q_{i,t}$, producing the top-$k$ scored GSW pairs $\{(q_j^G, a_j^G, s_j)\}_{j=1}^k$. To advance to hop $t + 1$, a GSW answer $a_{i,t}^G$ is selected to instantiate the next sub-question: $x_{i,t+1} = a_{i,t}^G$.

**Chains.** A *chain* $C$ is an ordered sequence of GSW QA pairs accumulated across hops for a sub-question sequence $i$. At each hop $t$, the sub-question $q_{i,t}(x_{i,t})$ retrieves a GSW QA pair $(q_{i,t}^G, a_{i,t}^G)$ from memory. A chain traces:

$$(q_{i,1}^G, a_{i,1}^G) \Rightarrow (q_{i,2}^G, a_{i,2}^G) \Rightarrow \cdots \Rightarrow (q_{i,T}^G, a_{i,T}^G),$$

where the answer $a_{i,t}^G$ instantiates the next sub-question in the QA-decomposed sequence: $x_{i,t+1} = a_{i,t}^G$.

$B$**-Chain Construction and Pruning.** Because cross-encoder reranking is a noisy approximation of true relevance, committing to a single top-scoring candidate at each hop risks propagating early errors through the entire chain. We therefore maintain $B$ chains in parallel, each selecting a different GSW answer from the candidate set. Given a chain $C_j$ at the $t^{th}$ step, we score it by the geometric mean of their constituent relevance scores:

$$\text{score}(C_t(j)) = \left( \prod_{l=1}^t s_{l,j} \right)^{1/t} \tag{1}$$

where $s_{l,j}$ is the relevance score of the GSW pair in chain $j$ selected at hop $l$. We empirically validate this scoring function against alternatives in Appendix E.1.4.

At the first hop $t = 1$, corresponding to the sub-question $q_{i,1}(x_{i,1})$ (*where $x_{i,1}$ is already specified in QA decomposition*), the RETRIEVEANDSCORE subroutine returns top-$k$ (sorted by $s_{i,1,j}$) candidate QA pairs $\{(q_{i,1,j}^G, a_{i,1,j}^G, s_{i,1,j})\}_{j=1}^k$. Out of this $k$, we retain the top-$B$ $\{(q_{i,1,j}^G, a_{i,1,j}^G, s_{i,1,j})\}_{j=1}^B$ where the $a_{i,1,j}^G$'s are unique. This comprises $B$-chains each of length 1 and score $s_{i,1,j}$.

At each subsequent hop $t > 1$, it receives $B$ chains from the previous step $(t - 1)$, with answers $\{a_{i,t-1,j}^G\}_{j=1}^B$ for the $(t - 1)^{th}$ step, and cumulative chain scores $\{s_{i,(t-1),j}^*\}_{j=1}^B$ as computed in Eqn. 1. We first initiate $B$ queries each with one of the answers from step $(t - 1)$, $\{q_{i,t}(a_{i,t-1,j}^G)\}_{j=1}^B$ and call RETRIEVEANDSCORE subroutine for each. Each such query (i.e. fixed $j$) results in $k$ QA pairs $\{(q_{i,t,l,j}^G, a_{i,t,l,j}^G, s_{i,t,l,j})\}_{l=1}^k$, where $s_{i,t,l,j}$ is the cross-encoder similarity score between $q_{i,t}(a_{i,t-1,j}^G)$ and $q_{i,t,l,j}^G$. To merge across chains, we weigh each chain's $k$ answers by its cumulative scores (see 1): $\{s_{i,t,l,j}^* = (s_{i,(t-1),j}^*)^{(t-1)} * s_{i,t,l,j}\}_{l=1}^k$. The resulting $k * B$ long-list (across all chains $j = 1, ..., B$) is pruned based on the

**Algorithm 1** Reasoning Inference Chain Retrieval (RICR)

**Input:** Sub-questions $\{q_{i,t}(x_{i,t})\}$, beam width $B$, candidates $k$
**Output:** Evidence set $\mathcal{E}$
1: $\mathcal{C} \leftarrow$ RETRIEVEANDSCORE$(q_{i,1}(x_{i,1}), k)$ {Hop 1}
2: Retain top-$B$ by score, one per unique GSW answer
3: **for** $t = 2, \ldots, T$ (**Hops** 2–$T$) **do**
4:     **for** each $C \in \mathcal{C}$ with current answer $a_{i,t-1}^G$ **do**
5:         Instantiate $q_{i,t}(a_{i,t-1}^G)$
6:         Extend $C$ with each candidate from RETRIEVEANDSCORE$(q_{i,t}, k)$
7:         Update score$(C) \leftarrow (\prod_{j=1}^t s_j)^{1/t}$ {Geometric mean}
8:     **end for**
9:     Retain top-$B$ by score, one per unique current answer
10: **end for**
11: $\mathcal{E} \leftarrow$ deduplicated GSW pairs from all $C \in \mathcal{C}$
12: **return** $\mathcal{E}$

cumulative scores, and only the top-$B$ lists with unique current answers are retained—if multiple chains propose the same answer $a_{i,t}^G$, only the highest-scoring chain survives. Each winning entry forms a new chain of length $t$: retain the chain up to $(t - 1)$ and add the last QA in the winning entry. Then update the cumulative score according to Eqn. 1.

This ensures diverse exploration across entity paths rather than redundant reasoning toward the same entity (we show in Appendix E.1.5 that even $B = 1$ performs competitively; $B > 1$ provides consistent but modest gains capped at about $B = 5$).

After all $T$ hops, the top-$B$ surviving chains are collected. Their GSW QA pairs—*not the original document chunks*—are deduplicated and passed to the answer model as evidence.

For example, given the query "When did Lothair II's mother die?" (Fig. 2), hop 1 retrieves candidates for "Who was the mother of Lothair II?", seeding $C_1$ with entity Ermengarde of Tours ($s_1 = 0.92$) and $C_2$ with Ermengarde of Hesbaye ($s_1 = 0.78$). At hop 2, $C_1$ instantiates "When did Ermengarde of Tours die?", retrieving "A: 20 March 851" ($s_2 = 0.94$); $C_2$ instantiates "Where did Ermengarde of Hesbaye die?", retrieving "A: Angers" ($s_2 = 0.93$). Chain scores update to $\sqrt{0.92 \times 0.94} = 0.93$ and $\sqrt{0.78 \times 0.93} = 0.85$ respectively. $C_1$ ranks higher—the geometric mean propagates the stronger first hop. The evidence from $C_1$ is passed to the answer model, which generates: 20 March 851. For queries with multiple parallel sub-question sequences, the same procedure executes independently for each sequence, and the resulting evidence is combined. An end-to-end example of the RICR and its chains can be found in Appendix C. The full procedure is summarized in Algorithm 1.

# 3. Experimental Setup

## 3.1. Datasets

We evaluate PANINI through the lens of non-parametric continual learning benchmarks that emphasize (i) *factual memory* (single-hop retrieval of a directly stated fact) and (ii) *associativity* (composing information across documents, where intermediate entities must be discovered and used to reach the final answer). We use single-hop and multi-hop QA as a controlled testbed to evaluate these abilities.

**Multi-hop benchmarks.** MuSiQue (Trivedi et al., 2022), 2WikiMultihopQA (Ho et al., 2020), HotpotQA (Yang et al., 2018), and LV-Eval (hotpotwikiqa-mixup 256k) (Yuan et al., 2024) test multi-step reasoning across documents. MuSiQue targets compositional reasoning requiring multiple hops; 2WikiMultihopQA emphasizes diverse multi-hop reasoning patterns over Wikipedia; HotpotQA requires identifying and reasoning over supporting facts from multiple sources.
**Single-hop benchmarks.** NQ (Kwiatkowski et al., 2019) and PopQA (Mallen et al., 2023) test simpler factual retrieval, verifying that structured memory maintains strong performance on straightforward queries.

For fair comparison across baselines, we use the same benchmark subsets/splits used in HippoRAG2 (Gutiérrez et al., 2025a). Table 1 summarizes the dataset statistics.

## 3.2. Platinum: Reliability Under Missing Evidence

To evaluate reliability as introduced in §1, we require a setting where some queries are *genuinely unanswerable* from the available corpus. Inspired by the "platinum benchmark" philosophy (Vendrow et al., 2025), we construct MUSIQUE-PLATINUM and 2WIKI-PLATINUM.

**Construction.** We use a multi-agent LLM system to produce answers given only the source documents, then manually review for inconsistencies and label each example as **answerable** or **unanswerable**. In practice, unanswerable examples arise from missing or insufficient evidence, annotation errors, or ambiguity that prevents a single evidence-backed answer. The resulting splits contain 766/153 (ans/unans) for MuSiQue and 906/94 for 2Wiki.

**Evaluation.** All methods use the same prompt instructing the model to answer from evidence or output N/A when insufficient. We report **Ans** (F1 on answerable subset) and **Unans** (binary refusal accuracy on unanswerable subset).

## 3.3. Metrics

**Performance.** Following practice, we report **Exact Match (EM)** and **F1** (Yang et al., 2018; Trivedi et al., 2022). EM measures string match after normalization. F1 computes token-level overlap between predicted and gold answers.

**Inference-time efficiency.** We report the average inference-time token usage for answering a question (measured as total prompt tokens provided to the answer model).

**Reliability under missing evidence.** We report **Ans** and **Unans** on our Platinum evaluation (§3.2), which separates answerable from unanswerable questions with respect to the available corpus evidence. Ans measures EM/F1 on the answerable subset; Unans measures refusal accuracy on the unanswerable subset (correct iff the model outputs the canonical non-answer token).

## 3.4. Baselines

We compare PANINI against non-parametric memory systems spanning (i) chunk-based retrieval, (ii) structure-augmented memories, and (iii) agentic multi-step retrieval.

**Chunk-based retrieval.** We include **BM25** (Robertson & Zaragoza, 2009) as a classical sparse baseline over text chunks, and **BM25 + reranker**. We also evaluate **dense retriever** based on strong opensource encoder models (Lee et al., 2025; Zhang et al., 2025), and also report **dense + reranker**. We pass top-5 documents to the answering LLM for all chunk based retrieval methods.

**Structure-augmented memory systems.** We compare against methods that invest additional structure at write time: **RAPTOR** (Sarthi et al., 2024) (hierarchical summaries), and **GraphRAG** (Edge et al., 2025). We also include **HippoRAG** (Gutiérrez et al., 2025a;b) which constructs a knowledge graph for retrieval via Personalized PageRank (Haveliwala, 2002), though it returns passage chunks rather than structure augmented summaries.

**Agentic multi-step retrieval.** To represent inference-time iterative "plan–retrieve–refine" strategies, we evaluate **IR-CoT** (Trivedi et al., 2023) and **Search-R1** (Jin et al., 2025) using **BM25** as the underlying retriever. For **Search-R1**, we use the official released agent model, a fine-tuned Qwen2.5-7B. We focus on BM25 for these agentic systems in the main paper to control retrieval strength and isolate the cost of iterative inference; additional agentic variants (including dense retrieval) are reported in Appendix E.3.2.

**Answer models and prompting.** We use **GPT-4o-mini** (OpenAI et al., 2024) as the default answer model across all baselines. We additionally report results using opensource models in Appendix D. Non-GSW baselines use standard evidence prompts from their respective implementations, while PANINI formats retrieved evidence as QA pairs. We include a direct no-retrieval GPT-4o-mini baseline in Appendix E.3.1 to separate answer-model capability from retrieval quality.

*Table 1.* Dataset statistics. We use the same benchmark splits as HippoRAG 2 (Gutiérrez et al., 2025a).

|          | NQ    | PopQA | MuSiQue | 2Wiki | HotpotQA | LV-Eval |
|----------|-------|-------|---------|-------|----------|---------|
| Queries  | 1,000 | 1,000 | 1,000   | 1,000 | 1,000    | 124     |
| Passages | 9,633 | 8,676 | 11,656  | 6,119 | 9,811    | 22,849  |

### 3.5. Implementation Details

We construct GSW structures using **GPT-4.1-mini** as described in §2.1, processing each document independently. Question decomposition uses **GPT-4o**; chain-following retrieval uses beam width $B = 5$ with BM25 over entities and dense retrieval (**Qwen3-8B**) over QA pairs, reranked by **VoyageAI Rerank-2.5**. We use **GPT-4o-mini** as the answer model; for Platinum, the model outputs `N/A` when evidence is insufficient. Full hyperparameters appear in Appendix A and a reproduction of the entire framework with open-source models is presented in Appendix D.

## 4. Results and Discussion

We evaluate PANINI along the three NPCL criteria defined in § 1. **QA performance and efficiency** (Table 2): PANINI achieves the best average (56.1) across six benchmarks, outperforming the strongest structure-augmented baseline HippoRAG2 (53.3) and dense retrieval methods (50.5). Gains are most pronounced on multi-hop tasks, where PANINI also outperforms agentic systems despite using only a single decomposition call. Simultaneously, PANINI uses $2.2\times$ fewer tokens than chunk retrieval and $5$–$30\times$ fewer than structure-augmented and agentic methods by conditioning the answer model only on short, targeted QA pairs. **Reliability under missing evidence** (Table 3): Most systems exhibit a trade-off—weak retrievers abstain often but miss answerable questions; strong retrievers answer more but hallucinate under insufficient evidence. PANINI breaks this trade-off, achieving the highest answerable accuracy (79.8) while maintaining strong refusal accuracy (74.0). We report a detailed analysis of computational costs in Appendix H, including write-time indexing costs (Table 16), normalized write and read time requirements along with a detailed comparison against strong existing baselines (Table 20).

**Open-source pipeline.** To assess accessibility without proprietary APIs, we evaluate PANINI with fully open-source components (full details in Appendix D). Tables 6–7 replace both question decomposition (LoRA-finetuned Qwen3-8B) and answer generation (Qwen3-4B/8B) with open-source models (Yang et al., 2025). The same fixed decomposer is trained only on MuSiQue training data and then applied unchanged across the evaluation suite; Appendix D.1 further reports results with prompted, no-finetuning decomposers. While absolute performance decreases, PANINI's advantage over baselines is preserved—and in fact widens on

multi-hop tasks, where HippoRAG2 degrades more substantially. Platinum results (Tables 8–9) show similar trends. Finally, Table 10 replaces GSW construction itself with open-source models (Qwen3-8B/14B, GPT-OSS-120B) (Yang et al., 2025; OpenAI et al., 2025), yielding a fully open pipeline where every component fits on a single GPU. *These experiments also serve as a robustness test:* smaller models produce noisier GSW extractions (occasional missing verb-phrases or incomplete QA pairs, see Appendix F.2 for a detailed breakdown) yet performance degrades gracefully, with GPT-OSS-120B *still outperforming all baselines in Table 2 with access to proprietary APIs*. This demonstrates that RICR's beam search over multiple chains provides resilience to extraction errors, and that PANINI's gains stem from the framework design rather than extraction fidelity.

**Ablations.** We run extensive ablation studies over all components of PANINI; full results appear in Appendix E, and we summarize the main takeaways here. Reducing beam width from 5 to 3 has minimal impact on accuracy while cutting token usage by about 25% (Table 12). Even a single-beam variant remains competitive, although a wider search is more helpful for multi-hop questions. Furthermore, we study the effect of chain-level scoring in beam search by comparing cumulative (geometric-mean) scoring against similarity-only, combined, and greedy last-hop selection variants (Table 13). We also test robustness under corpus growth by holding the relevant evidence fixed and progressively adding distractor documents; results are reported in Appendix E.2. Finally, we assess whether PANINI's structured memory and RICR retrieval transfer to other systems: swapping Search-R1's chunk retrieval with our GSW index, without any additional training, improves performance on 4 of 6 datasets (Appendix E.3.2). Appendix F complements these results with qualitative analysis, highlighting baseline failure modes that PANINI avoids and summarizing PANINI's remaining failure cases.

## 5. Related Work

**Parametric vs. non-parametric continual learning.** Continual learning for language models broadly divides into parametric and non-parametric approaches. Parametric methods update model weights via continual pre-training, instruction tuning, or test-time adaptation (Sun et al., 2020; Wang et al., 2021; Ke et al., 2023). While these can internalize new knowledge, they risk catastrophic forgetting (Kirkpatrick et al., 2017), require expensive retraining, and inter-

*Table 2.* Performance (F1 ↑) and token efficiency (Tokens ↓) across six QA benchmarks. PANINI achieves the highest average performance while using 2–30× fewer tokens than competitive baselines. **Bold** = best; underline = second best. Format: **F1 Score** (Tokens).

| Method | Simple QA | | Multi-Hop QA | | | | |
|---|---|---|---|---|---|---|---|
| | NQ | PopQA | MuSiQue | 2Wiki | HotpotQA | LV-Eval | Avg |
| *Sparse Retrieval* | | | | | | | |
| BM25 | 56.0 (702) | 53.0 (665) | 25.3 (569) | 35.4 (545) | 54.2 (650) | 7.8 (1100) | 38.6 (705) |
| BM25+rerank | 58.2 (702) | 56.8 (665) | 29.9 (569) | 44.1 (545) | 63.4 (650) | 8.1 (1100) | 43.4 (705) |
| *Dense Retrieval* | | | | | | | |
| Qwen3 | 59.1 (702) | 59.8 (665) | 39.4 (569) | 56.2 (545) | 69.2 (650) | 11.7 (1100) | 49.2 (705) |
| Qwen3+rerank | 61.4 (702) | **59.9** (665) | 43.7 (569) | 57.9 (545) | 68.2 (650) | 11.8 (1100) | 50.5 (705) |
| *Structure-Augmented RAG* | | | | | | | |
| RAPTOR | 54.5 (1162) | 55.1 (1101) | 39.2 (942) | 48.4 (901) | 64.7 (1076) | 9.2 (1820) | 45.2 (1167) |
| GraphRAG | 55.5 (8076) | 51.3 (7659) | 42.0 (6555) | 61.0 (6265) | 67.6 (7487) | 11.0 (12661) | 48.1 (8122) |
| HippoRAG 2 | 60.0 (702) | 55.7 (665) | 49.3 (569) | 69.7 (545) | 71.1 (650) | 14.0 (1100) | 53.3 (705) |
| *Agentic Systems* | | | | | | | |
| IRCoT | 26.4 (15864) | 53.8 (7583) | 44.2 (11573) | 64.9 (7330) | 64.3 (7629) | 9.1 (14492) | 43.8 (10745) |
| Search-R1 | 47.9 (2075) | 49.7 (2053) | 41.1 (2242) | 64.9 (2426) | 68.6 (1986) | 11.5 (3964) | 47.3 (2458) |
| PANINI | **67.4** (432) | 57.6 (459) | **52.3** (192) | **72.4** (315) | 71.9 (288) | **14.8** (232) | **56.1** (320) |

*Table 3.* Platinum dataset evaluation with GPT-4o-mini. Ans = answerable, Unans = unanswerable. **Bold** = best; underline = second best.

| Retrieval | MuSiQue Platinum | | 2Wiki Platinum | | Avg | |
|---|---|---|---|---|---|---|
| | Ans ↑ | Unans ↑ | Ans ↑ | Unans ↑ | Ans ↑ | Unans ↑ |
| *Sparse Retrieval* | | | | | | |
| BM25 | 35.0 | **78.4** | 37.9 | **79.8** | 36.4 | **79.1** |
| BM25 + reranker | 42.3 | 74.5 | 47.9 | 78.7 | 45.1 | 76.6 |
| *Dense Retrieval* | | | | | | |
| Qwen3 (8B) | 53.3 | 63.4 | 63.5 | 70.9 | 58.4 | 67.2 |
| Qwen3 + reranker | 59.7 | 62.8 | 63.8 | 67.7 | 61.7 | 65.2 |
| *Structure-Augmented RAG* | | | | | | |
| HippoRAG 2 | 63.6 | 50.3 | 81.5 | 66.7 | 72.5 | 58.5 |
| PANINI | **75.0** | 72.6 | **84.8** | 73.1 | **79.9** | 72.8 |

act poorly with post-training alignment (Luo et al., 2023). Related work in information retrieval studies evolving corpora through parametric or retriever-update mechanisms, including continual learning for generative retrieval (Chen et al., 2023) and detecting when dense retrievers should be updated (Ko et al., 2025). Non-parametric methods keep weights fixed and store information externally, with retrieval-augmented generation (RAG) being the dominant paradigm (Lewis et al., 2021). PANINI falls in this category, but differs from standard RAG by writing structured, reusable abstractions rather than storing retrievable text fragments. We provide an extended discussion of parametric and evolving-corpus retrieval approaches in Appendix G.1.

**Sleep-time compute and precomputation.** Recent work on *sleep-time compute* (Lin et al., 2025) shows that pre-processing context during idle periods—before queries arrive—can reduce inference costs while improving accuracy. PANINI embodies a similar philosophy: invest computation at write time to save at read time. However, while sleep-time compute produces enriched but unstructured context,

PANINI produces *structured* representations—entity-linked QA networks—that enable multi-hop retrieval via traversal and reranking without additional LLM calls. This allows PANINI to match the compositional reasoning of agentic systems without their expensive, iterative inference loops, effectively trading write-time compute for read-time efficiency. **Structured retrieval and knowledge graphs.** We discuss structured retrieval approaches (RAPTOR, GraphRAG) and knowledge-graph methods (HippoRAG) in §1 and Appendix G.3, G.4. Multi-hop QA benchmarks and agentic retrieval methods are discussed in Appendix G.2.

## 6. Concluding Remarks and Limitations

In this work we study non-parametric continual learning through external structured memory and propose three evaluation criteria for such systems—supported performance, inference-time efficiency, and reliable abstention. We then present PANINI, which achieves superior performance across all three compared to existing methods. These results

validate our thesis that structuring experiences at write time yields both efficiency and reliability gains at read time.

**Limitations and Future Work.** Our current design leaves several important directions open. First, PANINI does not yet perform *latent-link caching* or experience-driven reconciliation: if certain cross-document relations recur frequently at inference time, the memory could be updated to reconcile these links and further reduce future retrieval cost. Second, while we show that open-source pipelines are viable, constructing high-quality GSWs remains more expensive with proprietary models and less reliable with smaller open-source models; reducing write-time cost and improving robustness of extraction and QA construction are key practical goals. Third, while our lightweight design avoids heavy global reconciliation, richer reconciliation policies – such as using an agent to navigate the GSW and reconcile entities and links – could strengthen the underlying structure. Fourth, our experiments evaluate non-parametric memory growth rather than parametric forgetting or explicit knowledge conflicts; studying conflicting updates, entity evolution, and stale memories is important future work. Finally, beyond fact-centric QA, extending PANINI to narrative-heavy settings and multimodal domains (e.g. long-form or streaming video) is a natural next step, where evolving entities, spatio-temporal structure, and cross-modal events may benefit even more from structured memory.

## Impact Statement

This paper presents work whose goal is to advance the field of Machine Learning. There are many potential societal consequences of our work, none which we feel must be specifically highlighted here.

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

# Appendix

Our technical appendix is structured as follows:

1. Appendix A: Prompts to the LLM.

2. Appendix B: GSW Representation Example.

3. Appendix C: RICR Example.

4. Appendix D: Open-Source Experiments.

5. Appendix E: Ablation Studies.

6. Appendix F: Qualitative Analysis.

7. Appendix G: Additional Related Work.

8. Appendix H: Computational Costs and Resources.

# A. Prompts to the LLM

In this section, we describe the prompt used for GSW construction, Question Decomposition, and Answering Models.

## A.1. GSW Representation

We present the prompt used to generate the GSW representation in Figure 4. The full prompt is substantially longer and includes detailed examples for each instruction; for clarity and space considerations, we omit these examples in the figures and retain only the core instructional text.

## A.2. Question Decomposition

We present the prompt for the question decomposition models in Fig 6.

## A.3. Answering Model

We present the prompt for the answering models in Fig 7.

# B. GSW Representation Example

Figure 3 provides a concrete example of our per-document Generative Semantic Workspace (GSW) representation: given a raw passage (top), we extract (i) entity nodes annotated with roles and states, and (ii) verb-phrase nodes and (iii) question–answer pairs that connect verb phrases to entities.

# C. RICR Example

We present an example RICR trace for PANINI in Figure 8. We report the question decomposition, hop subroutine, chain construction and pruning, and the final evidence used for answer generation. The raw trace logs are substantially longer (including the full top-$k$ entity lists, all reranked QA pairs, and complete beam states); for clarity and space considerations, we omit these details in the figures and retain only the highest-ranked retrieved items and the surviving chains at each hop.

# D. Open-Source Experiments

To assess the accessibility and robustness of PANINI without reliance on proprietary models—which might be crucial for applications with strict data privacy requirements—we evaluate the framework using open-source components. We organize these experiments along two complementary axes corresponding to read-time and write-time operations. First, we replace the proprietary question decomposition and answer generation models with open-source alternatives, while keeping GSW construction fixed (using GPT-4.1-mini). Second, we analyze the impact of varying the open-source

model used for GSW construction, with read-time components held constant. Finally, we combine these settings to evaluate a fully open-source pipeline, where every component—from memory construction to final answering—runs on open-source models.

## D.1. Open-Source Models at Read Time

### D.1.1. EXPERIMENTAL SETUP AND IMPLEMENTATION DETAILS

**Question Decomposition.** In the open-source read-time configuration, we use **Qwen3-8B + LoRA** and **Qwen3-4B + LoRA** decomposition models fine-tuned on curated gold decompositions. **GPT-5** is used as a teacher model to generate high-quality decomposition annotations for training data construction. Decomposition is performed greedily (temperature = 0.0) with thinking disabled for Qwen3-8B + LoRA and Qwen3-4B + LoRA.

**GSW Construction.** We use **GPT-4.1-mini** as the primary proprietary model for GSW construction, consistent with the main experiments presented in Table 2.

**Embedding model.** For dense retrieval and embedding-based matching, we use **Qwen3-Embedding-8B** with FAISS indexing.

**Reranking model.** For reranking top-$k$ retrieved we use **VoyageAI Rerank-2.5**.

**Reader models.** We evaluate the following open-source models as readers without any finetuning or prompt optimization, using the same prompt we used with GPT-4o-mini for our main experiments.

- **Qwen3-4B**: Table 6 (thinking disabled).

- **Qwen3-8B**: Tables 7 presents supported performance and 8 presented performance on the platinum set.

- **Qwen3-8B (thinking mode)**: Table 9.

As seen across models, we note that while performance drops across the board for all evaluated methods, PANINI suffers the least in terms of supported performance loss as well as abstention rates as measured by our platinum evaluation.

### D.1.2. TRAINING AND GENERATION HYPERPARAMETERS

In this section we detail our training and inference setup of the trained question decomposition models.

**Question Decomposition: Data** Training data is derived from the MuSiQue training set and consists of 1,780 question decomposition examples. Examples are grouped by

hop structure following the MuSiQue taxonomy: 2-hop, 3-hop (3hop1, 3hop2), and 4-hop (4hop1, 4hop2, 4hop3). Table 5 summarizes the final dataset composition by hop type. Decompositions are generated by prompting GPT-5 with our decomposition prompt on MuSiQue training questions. All generated decompositions are manually reviewed, and incorrect, redundant, or overly decomposed outputs are filtered out to ensure correctness and minimality.

**Question Decomposition: Training** The Qwen3-4B and Qwen3-8B decomposition model is fine-tuned using a LoRA (Hu et al., 2021) adapter with the following configuration:

- LoRA rank = 256

- learning rate = $2 \times 10^{-4}$

- batch size = 8

- number of epochs = 3

**Question decomposition: Inference** Once trained, we use the question decomposer with temperature set to 0. To test whether performance depends on decomposition fine-tuning, we also evaluate prompted GPT-OSS decomposers on MuSiQue without any decomposition fine-tuning. GPT-OSS-120B as both decomposer and answerer reaches 55.30 F1, GPT-OSS-20B as both decomposer and answerer reaches 52.85 F1, and GPT-OSS-20B decomposition with GPT-4o-mini answering reaches 52.25 F1; all remain above HippoRAG 2 under the main answerer setting (49.3 F1). **Answer generation.** For Qwen-based readers, we evaluate both non-thinking and thinking-enabled decoding modes. Table 4 summarizes the decoding hyperparameters used for answer generation under non-thinking and thinking-enabled configurations.

*Table 4.* Decoding hyperparameters for answer generation with open-source reader models.

| Parameter | Non-thinking Mode | Thinking Mode |
|---|---|---|
| Temperature | 0.6 | 0.6 |
| Top-$p$ | 0.95 | 0.95 |
| Top-$k$ | 20 | 20 |
| Max tokens | 4096 | $12{,}288$ |
| Repetition penalty | – | 1.1 |
| Presence penalty | – | 0.3 |
| Frequency penalty | – | 0.3 |

### D.1.3. QA WITH OPEN-SOURCE READERS

We next evaluate performance on standard QA benchmarks without explicit unanswerable questions. Table 6 reports results using Qwen3-4B as the end-to-end answerer model across a mix of simple and multi-hop datasets, including

*Table 5.* Distribution of question decomposition training data by hop type.

| Hop Type | Number of Examples |
|---|---|
| 2-hop | 494 |
| 3hop1 | 441 |
| 3hop2 | 406 |
| 4hop1 | 225 |
| 4hop2 | 49 |
| 4hop3 | 165 |
| **Total** | **1,780** |

MuSiQue, 2Wiki, and HotpotQA. Despite the limited capacity of the reader, PANINI consistently outperforms lexical, dense, and graph-based retrieval baselines on multi-hop benchmarks.

Scaling the answerer model to Qwen3-8B improves absolute performance across all methods. As shown in Table 7, PANINI continues to deliver the strongest or near-strongest results on multi-hop datasets, achieving the highest scores on MuSiQue and 2Wiki. These results indicate that structured GSW construction and chain-following retrieval remain effective as answerer model capacity increases.

In particular, the Qwen3-8B configuration recovers most of the performance of the proprietary answerer model setting, demonstrating that PANINI can operate near closed-source performance levels using fully open-source components.

### D.1.4. PLATINUM BENCHMARK EVALUATION

We first evaluate read-time open-source pipelines on Platinum benchmarks, which explicitly include unanswerable questions. Performance is reported separately for answerable questions (**Ans**) and unanswerable questions (**Unans**), where higher unanswerable accuracy reflects improved abstention and lower hallucination rates.

Table 8 reports results on MuSiQue Platinum and 2Wiki Platinum using Qwen3-8B as the end-to-end answerer without thinking mode. PANINI achieves the strongest answerable accuracy across both datasets, substantially outperforming lexical, dense, and graph-based baselines. In particular, PANINI attains an average answerable accuracy of 75.60%, compared to 66.98% for the strongest competing baseline (HippoRAG 2).

Table 9 reports results with thinking mode enabled for the answerer model. While thinking mode improves absolute answerable accuracy across most methods, PANINI remains the top-performing approach. The relative ordering of methods remains largely unchanged, indicating that the gains of PANINI arise primarily from structured retrieval rather than reader-side reasoning traces.

*Table 6.* **Performance comparison across multi-hop QA benchmarks with Qwen3-4B (no-think mode).** Experiments were run on **GPT-4.1-mini**-generated GSW memories, using **Qwen3-4B (no-think mode)** as the reader model and **Qwen3-4B + LoRA** for question decomposition (open-source PANINI pipeline). **Bold** = best; underline = second best.

| Retrieval | MuSiQue ↑ | 2Wiki ↑ | HotpotQA ↑ | Avg ↑ |
|---|---|---|---|---|
| *Sparse Retrieval* | | | | |
| BM25 | 17.07 | 36.74 | 48.30 | 34.04 |
| BM25 + reranker | 23.16 | 43.78 | 57.67 | 41.54 |
| *Dense Retrieval* | | | | |
| Qwen3-Embedding (8B) | 40.14 | 56.50 | 69.47 | 55.37 |
| Qwen3-Embedding (8B) + reranker | 46.30 | 58.34 | **72.04** | 58.89 |
| *Structure-Augmented RAG* | | | | |
| HippoRAG 2 | 42.20 | 66.79 | 70.33 | 59.77 |
| PANINI | **49.46** | **69.09** | 69.50 | **62.68** |

*Table 7.* **Performance comparison across QA benchmarks with Qwen3-8B(no-think mode).** Experiments were run on **GPT-4.1-mini**-generated GSW memories, using **Qwen3-8B (no-think mode)** as the reader and **Qwen3-8B + LoRA** for question decomposition (open-source PANINI pipeline). **Bold** = best; underline = second best.

| Retrieval | Simple QA | | Multi-Hop QA | | | |
|---|---|---|---|---|---|---|
| | NQ ↑ | PopQA ↑ | MuSiQue ↑ | 2Wiki ↑ | HotpotQA ↑ | Avg ↑ |
| *Sparse Retrieval* | | | | | | |
| BM25 | 45.01 | 47.61 | 18.96 | 29.28 | 48.80 | 37.93 |
| BM25 + reranker | 53.57 | 51.27 | 24.34 | 38.61 | 59.95 | 45.55 |
| *Dense Retrieval* | | | | | | |
| Qwen3-Embedding (8B) | 59.90 | 59.74 | 39.97 | 55.66 | 69.26 | 56.91 |
| Qwen3-Embedding (8B) + reranker | 61.27 | 60.01 | 45.60 | 58.10 | 72.29 | 59.45 |
| *Structure-Augmented RAG* | | | | | | |
| HippoRAG 2 | 60.01 | 55.51 | 45.40 | 66.93 | **72.23** | 60.02 |
| PANINI | **65.31** | **57.21** | **52.43** | **70.28** | 72.14 | **63.47** |

*Table 8.* **Platinum dataset evaluation with Qwen3-8B (no-think mode).** **Ans** = F1 score on the *answerable* subset. **Unans** = refusal accuracy on the *unanswerable* subset (binary; correct iff the output is the canonical non-answer token N/A after normalization). Experiments were run on **GPT-4.1-mini**-generated GSW memories, using **Qwen3-8B (no-think mode)** as the reader and **Qwen3-8B + LoRA** for question decomposition (open-source PANINI pipeline). **Bold** = best; underline = second best.

| Retrieval | MuSiQue Platinum | | 2Wiki Platinum | | Avg | |
|---|---|---|---|---|---|---|
| | Ans ↑ | Unans ↑ | Ans ↑ | Unans ↑ | Ans ↑ | Unans ↑ |
| *Sparse Retrieval* | | | | | | |
| BM25 | 20.17 | **82.35** | 30.24 | **88.30** | 25.21 | **85.33** |
| BM25 + reranker | 27.79 | 76.47 | 40.37 | 83.33 | 34.08 | 79.90 |
| *Dense Retrieval* | | | | | | |
| Qwen3-Embedding (8B) | 40.88 | 66.01 | 49.02 | 75.18 | 44.95 | 70.60 |
| Qwen3-Embedding (8B) + reranker | 48.14 | 62.09 | 56.68 | 66.67 | 52.41 | 64.38 |
| *Structure-Augmented RAG* | | | | | | |
| HippoRAG 2 | 53.66 | 50.46 | 80.30 | 63.48 | 66.98 | 56.97 |
| PANINI | **68.08** | 55.56 | **83.11** | 67.02 | **75.60** | 61.29 |

Lexical baselines such as BM25 achieve high unanswerable accuracy by aggressively abstaining, but this behavior results in substantially lower answerable accuracy. In contrast, PANINI maintains competitive unanswerable performance while substantially improving answerable accuracy, indicating better calibration rather than reliance on excessive abstention. This balance is critical in Platinum settings, where unanswerable accuracy directly reflects hallucination control.

### D.2. End-to-end Open-Source Pipeline

#### D.2.1. EXPERIMENTAL SETUP AND IMPLEMENTATION DETAILS

To assess the robustness of GSW construction beyond large proprietary models, and to support settings where data governance or safety constraints require fully local deployment, we vary the *open-source* model used for GSW construction. Unless otherwise noted, all experiments in this subsection use a fully open-source read-time configuration: question decomposition is performed by a LoRA-finetuned **Qwen3-8B** decomposer, and answer generation uses **Qwen3-8B** in no-thinking mode. Full read-time model details and decoding/training hyperparameters follow Appendix D.1.

**Write-time GSW construction models.** We instantiate GSW construction using the following open-source LMs: **Qwen3-8B**, **Qwen3-14B**, and **GPT-OSS-120B**. We additionally evaluate a two-pass refinement variant for **Qwen3-14B**, where the model performs an initial GSW extraction followed by a self-refinement pass to repair missing entities, verb-phrase nodes, and malformed QA pairs.

#### D.2.2. RESULT ANALYSIS

Results on representative multi-hop QA benchmarks (MuSiQue and 2Wiki) are summarized in Table 10. We observe that GSW quality and downstream QA performance scale consistently with model capacity. Performance improves approximately monotonically as model size increases, with Qwen3-14B outperforming Qwen3-8B, and GPT-OSS-120B achieving the strongest overall performance, achieving competitive performance relative to the main PANINI configuration that uses proprietary models for GSW construction, question decomposition, and answer generation. These results indicate that both GSW construction and the overall PANINI pipeline remain effective under a fully open-source model configuration.

#### D.2.3. TWO-PASS GSW REFINEMENT

Further analysis suggests that the performance gap observed with smaller models arise primarily during GSW construction: smaller models occasionally omit critical verb-phrase nodes, fail to extract complete event / verb nodes, or gen-

erate incorrect or incomplete question–answer pairs. Such local structural and information omissions can propagate during chain-following retrieval, resulting in broken or truncated reasoning paths at inference time.

To mitigate these issues while preserving the cost advantages of smaller open-source models, we introduce a two-pass GSW construction strategy. In the first pass, the model generates an initial GSW representation from the document. In the second pass, the same model is prompted to explicitly inspect the partially constructed GSW and to repair structural deficiencies by adding missing entities or relations, and correcting malformed question–answer pairs. The prompt for the second pass models is shown in Figure 5.

As shown in Table 10, the second-pass refinement consistently improves downstream performance. This shows that a smaller open-source constructor can reach strong GSW quality when paired with a lightweight multi-step refinement pass, leading to competitive downstream performance.

## E. Ablation Studies

In this section, we analyze the key design choices of both PANINI and competitive baselines. First, we conduct ablation studies on the core components of our Reasoning Inference Chain Retrieval (RICR) procedure. Second, we evaluate agentic retrieval variants to benchmark PANINI's single-pass retrieval design against iterative read-time reasoning frameworks.

### E.1. RICR Design Ablation Studies

#### E.1.1. ABLATION: QUESTION DECOMPOSITION

We ablate question decomposition, which converts a multi-hop question into an ordered sequence of single-hop sub-questions prior to retrieval. In the full system, retrieved intermediate answers are injected into later hops using placeholders (e.g., `<ENTITY_Q1>`) to enforce chain-following retrieval. In the no decomposition variant, we bypass decomposition and issue a single retrieval step for the original question, followed by answer generation from the retrieved evidence. Table 11 reports the resulting performance. Disabling decomposition substantially reduces accuracy, showing that explicitly exposing intermediate hops is important for retrieving the correct bridging evidence.

#### E.1.2. ABLATION: DUAL SEARCH

We ablate the dual-search retrieval strategy used to propose candidates at each hop. In the full system, PANINI retrieves (i) entity candidates and (ii) QA-pair candidates, and combines them before reranking and beam expansion. In the **entity-only** variant, we disable QA pair retrieval and rely solely on entity retrieval to generate evidence; in the

**QA-only** variant, we disable entity retrieval and rely only on retrieved QA pairs. Table 11 reports the resulting performance. Dual search improves robustness by providing complementary candidate sources: entity-only retrieval is weaker, while QA-only retrieval is closer to the full system but still slightly underperforms in multi-hop settings.

### E.1.3. ABLATION: QA RERANKING

We ablate the QA reranking stage used to select the top-$k$ evidence QA pairs after retrieval. In the full system, we retrieve a candidate pool of QA pairs and rerank them with Voyage rerank-2.5, retaining the top-$k$ for beam expansion and final answer generation. In the no QA reranking variant, we skip reranking and instead keep the top-$k$ QA pairs in the original retrieval order. Table 11 reports the resulting performance. Removing reranking substantially degrades multi-hop performance, indicating that accurate evidence prioritization is critical for maintaining high-quality beams.

### E.1.4. ABLATION: CHAIN-LEVEL SCORING

We ablate the chain-level scoring function used to rank beams during multi-hop retrieval. Let $\mathcal{C} = \{(q_i, a_i)\}_{i=1}^n$ denote a reasoning chain and let $s_i$ be the Voyage reranker score for hop $i$ (Voyage rerank-2.5). Our **main** scoring rule aggregates hop-wise quality using the geometric mean,

$$S_{\text{cum}}(\mathcal{C}) = \left( \prod_{i=1}^n s_i \right)^{1/n},$$

which strongly penalizes chains containing any weak hop. As alternatives, we evaluate a **similarity-only** score based on cosine similarity between the embeddings of the original question $Q_{\text{orig}}$ and the linearized chain $L(\mathcal{C})$,

$$S_{\text{sim}}(\mathcal{C}) = \cos\big(\text{Embed}(Q_{\text{orig}}), \text{Embed}(L(\mathcal{C}))\big)$$

and a **combined** score that linearly interpolates local and global signals,

$$S_{\text{comb}}(\mathcal{C}) = \alpha S_{\text{cum}}(\mathcal{C}) + (1 - \alpha) S_{\text{sim}}(\mathcal{C}), \quad \alpha = 0.5.$$

Finally, **none (greedy)** disables chain-level scoring and ranks beams using only the most recent hop score (i.e., last-hop reranker score $s_n$ at each step). Table 13 reports the resulting performance. The main cumulative scoring rule performs best, while the similarity-only and combined variants underperform, suggesting that prioritizing consistently strong hop-level evidence is important for effective beam search; greedy last-hop selection is competitive but still falls short of the cumulative rule.

### E.1.5. ABLATION: BEAM WIDTH

We ablate the beam width $B$ used during chain-following retrieval, i.e., the maximum number of partial chains retained after pruning at each hop. We compare the default

setting ($B=5$) to wider search ($B=10$) and narrower search ($B=3$ and $B=1$), keeping all other hyperparameters fixed. Table 12 reports both F1 and average token usage over 1,000 questions. Reducing the beam from 5 to 3 preserves accuracy while substantially reducing token usage, whereas a single-beam setting ($B=1$) reduces token usage further but degrades performance, especially on multi-hop questions.

### E.2. Continual Learning Ablation Studies

As retrieval-augmented QA systems are deployed in real-world settings, they must remain reliable as the underlying corpus continuously grows. To isolate the effect of corpus expansion, we design a controlled robustness experiment on MuSiQue where we fix an evaluation set of 200 questions and progressively enlarge the retrieval corpus from 4K passages to the full collection ($\sim$12K). Importantly, all gold-supporting passages for these questions are already present in the initial 4K subset; thus, the number of relevant documents remains constant while the corpus grows only by adding *distractor* passages. This setup directly probes robustness to an increasing search space (and potential retrieval noise) without changing the underlying evidence needed to answer the questions.

Figure 9 reports F1 as the corpus expands. We observe that methods relying on lightweight structured memory, PANINI, degrade substantially less under expansion, maintaining more stable performance as distractors accumulate, whereas embedding- and BM25-based baselines exhibit larger drops as the retrieval problem becomes increasingly confounded by irrelevant content. These results suggest that explicit structure can improve resilience to distribution shifts induced purely by corpus growth, a common failure mode in continually evolving knowledge bases.

We additionally run a staged document-arrival protocol on 100 MuSiQue questions split into 10 groups. At each stage, we append the supporting documents for one additional group while keeping distractors present, then evaluate all 100 questions. Overall F1 rises from 22.7 to 82.9 as supporting evidence becomes available, while answerable-only F1 remains approximately flat at 83% across stages; the strongest baseline reaches 66.4 F1 at the final stage. This protocol tests incremental memory updates under document arrival, while leaving explicit knowledge conflicts and forgetting for future work.

### E.3. Additional Experiments

#### E.3.1. CONTROLLED RETRIEVAL AND DIRECT ANSWERING BASELINES

To separate retrieval quality from evidence formatting and answer-model strength, we include two additional controls. First, Table 14 compares RICR against direct dense retrieval

over the same GSW QA pairs used by PANINI; direct QA-vector retrieval performs substantially worse, showing that the gains are not due to the compact QA evidence format alone. Second, Table 15 reports a direct GPT-4o-mini baseline with no retrieved context.

### E.3.2. AGENTIC VARIANTS

Table 17 reports additional variants of **Search-R1** that modify the underlying retrieval method while keeping the rest settings fixed. These experiments are intended to disentangle the impact of retrieval design from the agentic reasoning loop itself.

**Search-R1 (Dense)** replaces BM25 with a dense retriever using `Qwen3-Embedding-8B`, the same embedding model used in PANINI. This variant consistently improves performance on multi-hop benchmarks, most notably on MuSiQue, HotpotQA, and LV-Eval.

**Search-R1 (Dense + QA)** further replaces document chunks with PANINI's extracted question–answer (QA) pairs as the retrieval corpus. This change yields competitive results on multi-hop datasets, suggesting that structured QA pairs can serve as effective representations even when used within an agentic framework originally designed for passage-level retrieval.

Finally, **Search-R1 (Dense + QA + Retrieval)** integrates PANINI's dual-indexing strategy, combining entity-level retrieval with direct QA-pair similarity search. This variant achieves the strongest overall performance among all Search-R1 configurations, with consistent gains across multi-hop benchmarks.

Across all variants, dense retrieval generally outperforms BM25, confirming that embedding-based retrieval provides stronger semantic matching for agentic multi-step QA. Moreover, the best-performing configuration, which is using PANINI's dual-index retrieval design, demonstrates that PANINI's retrieval design choices remains effective even when embedded inside an agentic reasoning system.

## F. Qualitative Analysis

In this section, we present qualitative analyses with representative examples that illustrate why PANINI outperforms competitive baselines on multi-hop reasoning tasks. We also analyze common failure modes to better understand the limitations of the method.

### F.1. Qualitative Analysis of PANINI Performance

We analyze representative failures of two strong non-parametric baselines: (i) dense embedding retrieval using Qwen3-Embedding-8B and (ii) graph-based retrieval via HippoRAG2. Table 18 provides concrete examples. Below

we summarize the dominant failure modes of these methods and explain how PANINI's explicit chain-following retrieval and structured memory design avoid these issues.

**Embedding retrieval failures (Qwen3-Embedding-8B).** A common failure pattern is *missing the second hop*: dense retrieval often over-matches the surface form of the first-hop entity and returns many semantically adjacent passages, but fails to recover the specific bridging evidence needed to instantiate the follow-up hop. PANINI mitigates this by performing explicit query decomposition and RICR: each hop is queried with an evolving context that incorporates previous hop answers and constraints, narrowing the retrieval target from broadly surface-level semantic similarity to hop- and chain-specific evidence.

We also observe *missing intermediate entities* in longer chains, where a single dense query cannot reliably retrieve all required supports (often spanning multiple documents and entity aliases). PANINI's GSW representation explicitly stores entity nodes with roles/states, improving recall for the entities with aliases; dual indexing further increases coverage by allowing retrieval to follow the chain from either an entity mention or a supporting QA.

Finally, *broken chains* occur when an answer-bearing passage is retrieved in isolation, without the supporting evidence that connects it to earlier hops, leaving the overall reasoning unsupported. PANINI avoids this failure by explicitly constructing and maintaining the reasoning chain during retrieval: each hop is selected as part of a growing, structured evidence path, rather than as an independent relevance match.

**Graph-based retrieval failures (HippoRAG2).** HippoRAG2 frequently exhibits *retrieval drift*: Personalized PageRank propagates relevance through loosely informative graph connections, promoting entities that are topically related but incorrect (e.g., location from the same city, or shared theme). As a result, the walk can over-rank "nearby" distractors while suppressing the specific entity needed to complete the multi-hop composition. PANINI mitigates this by avoiding query-agnostic graph propagation in favor of explicit, hop-conditioned retrieval. Through query decomposition and RICR, each hop is queried using the context accumulated from previous hops, so retrieval is guided by the current reasoning state rather than by graph proximity or neighborhood structure.

We also observe cases where HippoRAG2 over-focuses on the seed entity neighborhood, yielding *missing the second hop* evidence: the walk remains trapped in a locally coherent region when the correct bridge requires stepping outside that subgraph. PANINI reduces this failure mode by explicitly constructing the reasoning chain during retrieval: each

step updates the retrieval context using the newly identified intermediate entity and constraints, enabling the next hop to "move" to the correct evidence source instead of remaining confined to the initial neighborhood.

### F.2. Qualitative Error Analysis

We qualitatively analyze representative PANINI failure cases across datasets to better understand current limitations. We observe three recurring error modes: (i) missing verb-phrase nodes during GSW construction, (ii) QA-pair construction errors that omit necessary inverse links, and (iii) imperfect question decomposition for complex queries. Table 19 provides representative examples.

**Missing verb-phrase nodes.** Errors can arise during GSW construction when a key event or verb-phrase node is omitted. In this case, the relevant relation is only partially captured, leaving no QA pairs that support downstream chain-following retrieval (Table 19, Row 1).

**QA-pair construction errors.** We also observe failures caused by imperfect QA-pair construction, where an entity that should serve as an answer never appears as the answer to any question in the appropriate direction. Since later sub-questions are instantiated from entities retrieved by earlier QA pairs, such omissions can break the reasoning chain (Table 19, Row 2).

**Imperfect question decomposition.** Finally, some errors stem from imperfect question decomposition, particularly for long or ambiguity-prone queries. In these cases, the decomposer may generate sub-questions that do not align with the structure of the available evidence, causing downstream retrieval to follow an incorrect chain (Table 19, Row 3).

## G. Additional Related Work

This section expands discussions deferred from §6 (Related Work).

### G.1. Continual Learning

**Test-Time Training.** Test-Time Training (TTT) and Test-Time Adaptation (TTA) can be viewed as parametric continual learning approaches that operate during inference or deployment. Recent works explore test-time parameter adaptation for both vision models and large language models, where model updates are driven by objectives that do not require access to ground-truth task labels, such as next-token prediction, perplexity minimization, or auxiliary self-supervised losses computed on test-time or retrieved inputs (Sun et al., 2020; Wang et al., 2021; 2022; Hardt & Sun, 2024). By enabling label-free adaptation under distribution shift, TTT naturally supports continual learning in

non-stationary environments. However, because adaptation is guided solely by self-supervised signals, these methods may reinforce incorrect predictions, leading to error accumulation or catastrophic forgetting over time. In addition, performing gradient-based updates during inference introduces additional computational overhead and increases serving latency, which can limit practical deployment (Niu et al., 2023; Zhao et al., 2023).

**Continual fine-tuning.** Instead of adapting model parameters at test time using unlabeled inputs, a more traditional approach to continual learning is to *continually fine-tune* the model on a stream of new data or tasks in a supervised manner. To mitigate catastrophic forgetting and preserve prior capabilities, previous works have explored a range of mechanisms, including data replay or rehearsal (Lopez-Paz & Ranzato, 2017; Rolnick et al., 2019), regularization and constraints on parameter updates (Kirkpatrick et al., 2017; Li & Hoiem, 2017; Zenke et al., 2017), continual or domain-adaptive pre-training (Ke et al., 2023; Jin et al., 2022), as well as parameter-efficient adaptation techniques (Jin et al., 2022; Wang et al., 2023). Thus, continual fine-tuning can also be viewed as a form of *parametric continual learning*, as it explicitly updates model weights to integrate new knowledge over time. However, despite its conceptual simplicity, this paradigm faces several practical challenges. In particular, maintaining replay buffers can be memory- and data-intensive, strong regularization may limit model generalizability, and repeated fine-tuning causes substantial computational and operational cost, especially for large-scale models.

**In-context learning.** In-context learning (ICL) enables large language models to adapt to new tasks or distributions by conditioning on examples or instructions provided in the input context, without explicit parameter updates (Brown et al., 2020). As such, ICL can be viewed as a form of *non-parametric continual learning* and serves as the core mechanism underlying a broad class of prompt-based and RAG-based systems. However, in its standard form, in-context learning does not prescribe how contextual information is selected, stored, or accumulated over time.

**Evolving corpora in information retrieval.** Continual and dynamic-corpus retrieval work studies update policies for retrieval systems. Chen et al. (Chen et al., 2023) study continual learning for generative retrieval, where a parametric retriever incrementally absorbs new documents while preserving access to older ones. Ko et al. (Ko et al., 2025) instead ask when a dense retriever should be updated as a corpus evolves, using out-of-distribution detection before indexing new documents. These settings are complementary to PANINI: we keep the base model and retriever fixed and incorporate new information by writing documents into an external structured memory.

### G.2. Multi-Hop QA and Agentic Retrieval

Multi-hop question answering requires systems to connect information across multiple text passages through intermediate reasoning steps. Benchmarks such as HotpotQA (Yang et al., 2018), 2WikiMultihopQA (Ho et al., 2020), and MuSiQue (Trivedi et al., 2022) test this capability. The core challenge lies in bridging associative gaps: relevant information may exist in the corpus, but standard retrieval methods fail because query entities do not co-occur with answers in individual documents.

**Agentic approaches** address this through iterative LLM-based reasoning. IRCoT (Trivedi et al., 2023) interleaves chain-of-thought reasoning with retrieval, allowing the LLM to formulate follow-up queries based on intermediate answers. Self-Ask (Press et al., 2023) and ReAct (Yao et al., 2022) follow similar patterns of decomposing questions and retrieving iteratively. While effective, these methods require multiple LLM calls per question, leading to high token usage and latency. Recent work like Search-R1 (Jin et al., 2025) achieves better efficiency through specialized training for search-augmented reasoning, but requires additional training infrastructure. These approaches demonstrate the power of agentic reasoning but also highlight the computational costs of placing intelligence burden on the retrieval process.

These lines of work motivate the query-time cost of placing intelligence in the retrieval loop. In contrast, our approach shifts computation to write-time structure and performs multi-hop composition via lightweight chain retrieval with minimal query-time LLM usage.

### G.3. Additional Structured Representations for Retrieval

Other structured approaches include hierarchical document summarization methods like RAPTOR, which build tree structures of summaries at multiple levels of abstraction, and various forms of schema-based or template-based information extraction. These methods trade off between retrieval precision and coverage, often requiring either broad context windows or multiple retrieval rounds.

### G.4. Knowledge Graph-Based Approaches

Knowledge graph methods explicitly construct entity-relationship networks from text corpora, and recent surveys cover many ways of combining LLMs and KGs for QA (Ma et al., 2025). HippoRAG (Gutiérrez et al., 2025a) builds a knowledge graph connecting all entities across documents with weighted edges, then uses Personalized PageRank (PPR) to traverse this graph during retrieval. While PPR effectively spreads activation across connected entities, this approach has two key limitations. First, it creates

dense connectivity: entities that co-occur in text are linked regardless of semantic relatedness, leading to spurious connections (e.g., all dates mentioned in a corpus become interconnected). Second, PPR operates on the graph structure without fully leveraging the semantic reasoning capabilities of language models—the algorithm propagates activation based on edge weights but does not reason about the meaning of the connections (connection scores are based solely on dense vector similarity)

Our approach differs in that we avoid constructing a global graph with explicit entity links. Instead, we maintain separate structured representations for each document and use language model-guided beam search to dynamically form entity chains during retrieval. This design avoids precomputing potentially spurious connections while enabling semantic reasoning at each step of the chain.

## H. Computational Costs and Resources for Building the GSW

The primary computational costs of PANINI arise from the one-time *write-time* indexing step that constructs per-document GSW memories. We orchestrate large-scale parallel indexing calls using the Bespoke Curator library (Marten et al., 2025). For MuSiQue (11,656 passages), we ran the full GSW indexing pipeline with `gpt-4.1-mini` **using the OpenAI Batch API** and computed the total write-time cost from recorded token usage and pricing, yielding a one-time construction cost of **$48.02**. We summarize write-time indexing costs in Table 16. The serialized GSW memory bank for the same corpus occupies approximately 164 MB on disk in a human-readable JSON format, including entity nodes, verb-phrase/event nodes, and QA pairs. This storage cost scales linearly with the number of documents because PANINI stores document-local memories rather than a dense global cross-document graph; compact binary serialization would reduce this footprint further.

Table 20 summarizes end-to-end runtime and resource requirements on MuSiQue (11,656 passages). For PANINI and baselines, we report indexing time and QA latency per query. Importantly, all QA latency per query measurements were obtained using standard (non-Batch) API calls to reflect typical synchronous execution. Overall, PANINI shifts most of its compute to a one-time write-time indexing step (GSW construction); after indexing, read-time inference remains lightweight and is dominated by retrieval and reranking rather than multi-step LLM processing.

In addition, question decomposition is inexpensive in practice, requiring ~60 tokens on average per query and thus contributing negligibly to per-question runtime and cost. When new documents arrive, only their new GSW memories and embeddings need to be added; existing memories are un-

touched. On MuSiQue, the average write-time cost is about 0.515 seconds per passage, and refreshing the lightweight BM25 entity index takes under two seconds at this scale.

---

### Input passage (verbatim)

---

**Title:** *Lothair II*
**Text:** Lothair II (835 – ) was the king of Lotharingia from 855 until his death. He was the second son of Emperor Lothair I and Ermengarde of Tours. He was married to Teutberga (died 875), daughter of Boso the Elder.

### GSW Output

---

**Entities (role → states).**

- **Lothair II**: person → {deceased, historical figure}; ruler → {king of Lotharingia, medieval period}

- **835**: date → {birth year, medieval period}

- **Lotharingia**: location → {kingdom, medieval period}

- **855**: date → {start of reign, medieval period}

- **Emperor Lothair I**: person → {historical figure}; ruler → {Carolingian Emperor, medieval period}

- **Ermengarde of Tours**: person → {historical figure}; nobility → {medieval period}

- **Teutberga**: person → {deceased, historical figure}; nobility → {medieval period}

- **875**: date → {death year, medieval period}

- **Boso the Elder**: person → {historical figure}; nobility → {medieval period}

| Verb phrase node | Bidirectional QA pairs (with roles/states on answers) |
|---|---|
| **king of** | • **Q:** Who was the king of Lotharingia?
  **A: Lothair II**  (*person* → {deceased, historical figure}; *ruler* → {king of Lotharingia, medieval period})

• **Q:** What was Lothair II the king of?
  **A: Lotharingia**  (*location* → {kingdom, medieval period}) |
| **reigned from** | • **Q:** When did Lothair II start his reign?
  **A: 855**  (*date* → {start of reign, medieval period})

• **Q:** Who started reigning in 855?
  **A: Lothair II**  (*person* → {deceased, historical figure}; *ruler* → {king of Lotharingia, medieval period}) |
| **son of** | • **Q:** Who is the son of Emperor Lothair I?
  **A: Lothair II**  (*person* → {deceased, historical figure}; *ruler* → {king of Lotharingia, medieval period})

• **Q:** Who is Lothair II the son of?
  **A: Emperor Lothair I**  (*person* → {historical figure}; *ruler* → {Carolingian Emperor, medieval period}) |
| **son of** | • **Q:** Who is the son of Ermengarde of Tours?
  **A: Lothair II**  (*person* → {deceased, historical figure}; *ruler* → {king of Lotharingia, medieval period})

• **Q:** Who is Lothair II the son of?
  **A: Ermengarde of Tours**  (*person* → {historical figure}; *nobility* → {medieval period}) |
| **married to** | • **Q:** Who was Lothair II married to?
  **A: Teutberga**  (*person* → {deceased, historical figure}; *nobility* → {medieval period})

• **Q:** Who was married to Teutberga?
  **A: Lothair II**  (*person* → {deceased, historical figure}; *ruler* → {king of Lotharingia, medieval period}) |
| **daughter of** | • **Q:** Who is the daughter of Boso the Elder?
  **A: Teutberga**  (*person* → {deceased, historical figure}; *nobility* → {medieval period})

• **Q:** Who is Teutberga the daughter of?
  **A: Boso the Elder**  (*person* → {historical figure}; *nobility* → {medieval period}) |

*Figure 3.* **Example of a per-document Generative Semantic Workspace (GSW).** Top: the raw input passage (title + text). Bottom: the corresponding GSW, rendered as (i) entity nodes annotated with roles and states, and (ii) verb-phrase nodes instantiated as *bidirectional* question–answer (QA) pairs.

*Table 9.* **Platinum dataset evaluation with Qwen3-8B (thinking mode). Ans** = F1 score on the *answerable* subset. **Unans** = refusal accuracy on the *unanswerable* subset (binary; correct iff the output is the canonical non-answer token `N/A` after normalization). Experiments were run on **GPT-4.1-mini**-generated GSW memories, using **Qwen3-8B (thinking mode)** as the reader and **Qwen3-8B + LoRA** for question decomposition (open-source PANINI pipeline). **Bold** = best; underline = second best.

| Retrieval | MuSiQue Platinum | | 2Wiki Platinum | | Avg | |
|---|---|---|---|---|---|---|
| | Ans ↑ | Unans ↑ | Ans ↑ | Unans ↑ | Ans ↑ | Unans ↑ |
| *Sparse Retrieval* | | | | | | |
| BM25 | 18.04 | **89.54** | 28.62 | **93.97** | 23.33 | **91.76** |
| BM25 + reranker | 21.71 | 86.93 | 36.61 | 87.23 | 29.16 | 87.08 |
| *Dense Retrieval* | | | | | | |
| Qwen3-Embedding (8B) | 36.89 | 82.35 | 47.13 | 74.47 | 42.01 | 78.41 |
| Qwen3-Embedding (8B) + reranker | 41.26 | 73.20 | 49.81 | 76.60 | 45.54 | 74.90 |
| *Structure-Augmented RAG* | | | | | | |
| HippoRAG 2 | 51.19 | 70.74 | 64.16 | 69.15 | 57.68 | 69.95 |
| PANINI | **69.64** | 73.20 | **82.78** | 76.95 | **76.21** | 75.08 |

***SYSTEM PROMPT:***
*You are an expert linguist focused on extracting factual relationships and attributes from Wikipedia-style content. Your primary task is to analyze text to create structured semantic networks that capture key factual information such as dates, places, nationalities, and other attributes needed for multi-hop question answering.*
***USER PROMPT:***
*Given the following text, extract factual relationships and attributes following this structure:*

```
<input_text>
{input_text}
</input_text>
```

*Per-Rule Micro-Examples*
 *1) Atomic entities*
 *- Input: "Ubisoft's Assassin's Creed was announced on 29 October 1923 via New York Times with the request of US government."*
 *- Do: Entities "Ubisoft" (organization), "Assassin's Creed" (title/work), "29 October 1923" (date), "New York Times" (media), "US government" (government). Phrase: "announced".*
 *- Don't: Single entity "Ubisoft's Assassin's Creed". DO NOT pass any entity can be bundled into a single entity.*

 *2) Temporal rules (no fabrication)*
 *- Input: "On 29 October 1923, Parliament passed the Green Act."*
 *- Do: Entity "29 October 1923". Connect via an event/action: <Green Act> — "occurred during" → "29 October 1923".*

 *3) Abbreviation & alias*
 *- Input: "German Aerospace Center (DLR) led the study."*
 *- Do: Entities "German Aerospace Center (DLR)" (org) and "DLR" (alias); phrase "also known as".*
 *- Don't: Expand "UN" to "United Nations" if only "UN" appears.*

 *4) Two questions + complete content and recipient*
 *- Input: "Finance Minister Harald Jensen announced to Parliament that the budget would increase on May 19, 1919."*
 *- Do: Create proposition "the budget would increase"; add phrases with exactly one unknown per question:*
 *  - "announced to ... that" (recipient is the unknown):*
 *    - A→B: To whom did Finance Minister Harald Jensen announce \*\*that the budget would increase on May 19, 1919\*\*?*
 *    - B→A: Who announced to Parliament \*\*that the budget would increase on May 19, 1919\*\*?*
 *  - "announced that ... to" (content is the unknown):*
 *    - A→B: What did Finance Minister Harald Jensen announce to Parliament on May 19, 1919?*
 *    - B→A: Who announced \*\*that the budget would increase on May 19, 1919\*\*?*
 *  - "announced on ... to ... that" (date/time is the unknown):*
 *    - A→B: When did Finance Minister Harald Jensen announce to Parliament \*\*that the budget would increase\*\*?*
 *    - B→A: What did Finance Minister Harald Jensen announce to Parliament \*\*on May 19, 1919\*\*?*
 *  - \*\*DO NOT\*\* omit the \*\*that\*\* content in any question.*

 *5) Question format + IDs only*
 *- Input: "On 29 October 1923, Parliament passed the Green Act."*
 *- Do: "passed": A→B "Who passed the Green Act on 29 October 1923?" / B→A "What did Parliament pass on 29 October 1923?"; answers are IDs.*
 *- Don't: "Who passed it?" or answers with names.*

 *6) Universal object/content capture*
 *- Input: "The architect built the museum."*
 *- Do: "built" (agent→object) and "built by" (object→agent).*
 *- Input: "The minister reported to Cabinet that taxes would rise."*
 *- Do: "reported to" (recipient) and "reported that" (proposition).*

 *7) Authority/leadership context*
 *- Input: "Parliament passed the Green Act over President Henry Wallace's veto."*
 *- Do: "over veto of"; and "in office during".*

 *8) Special relationships (conditionality, purpose, temporal, comparative)*
 *- Conditional: "... would join NATO if forces were reorganized." → "conditional on".*
 *- Purpose: "... launched reforms to reduce inflation." → proposition "reduce inflation"; "for purpose".*
 *- Temporal: "... after elections." → temporal qualifier "after".*

 *9) Complete content capture*
 *- Input: "The CEO announced that the company would expand operations after securing funding."*
 *- Do: Proposition includes "after securing funding".*

 *10) Entity completeness*
 *- Input: "Congress voted on the controversial tax reform bill."*
 *- Do: Entity "controversial tax reform bill" so answers can reference it by ID.*

*Figure 4.* Prompt used for factual GSW construction from documents.

*11) Mandatory connectivity*
*- Input: "On March 15, 2020, the policy was announced."*
*- Do: Ensure the date appears in answers via a temporal phrase like "announced on".*

*12) Document Title Capturing*
*When a document title represents an alternative name for an entity:*
*- Put the title in parentheses after the primary entity name*
*- Create a separate entity node for the title itself*
*- Connect them with a "known as" relationship*
*- Input: "Ivan the Terrible \n Ivan IV was a historical figure who ruled Russia."*
*- Do: Create entity "Ivan IV (Ivan the Terrible)" and separate entity "Ivan the Terrible", then connect them with "known as" relationship.*
*- Output: Ivan IV (Ivan the Terrible) -> ruled -> Russia (location) , Ivan IV (Ivan the Terrible) -> known as -> Ivan the Terrible*

***Follow these examples for the desired extraction pattern:***

   ***(Examples omitted here for space; see source prompt in code.)***

***Key Instructions:***

  *1. **Extract ALL entities**: Include people, places, dates, titles, nationalities, professions, etc.*
   *- Do not encode answer-bearing values only as states; ensure a corresponding entity node exists (e.g., dates/times, locations, numbers/ordinals, titles/works, organizations, concepts) so When/Where/Who/What questions can reference them by ID.*

  *2. **Create relationship phrases**: These can be:*
   *- Factual attributes: "born on", "died on", "nationality", "profession"*
   *- Relationships: "directed by", "married to", "daughter of", "member of"*
   *- Properties: "English title", "released in", "located in"*

  *3. **Generate bidirectional questions**: Always create questions from both directions:*
   *- "Who was born in X?" AND "Where was Y born?"*
   *- "Who directed X?" AND "What did Y direct?"*
   *- QA inversion: If the A→B question contains entity A and its answers are the IDs of entity B, then the B→A question must contain entity B and its answers must be the IDs of entity A. The two questions must swap sides; do not repeat the same side in both questions' texts or answers.*

  *4. **No pronouns in questions or uncertain objects in questions.***
   *- Input: "Michael Jackson stated that his verses are about the convict life in Brasil in his song Care About US"*
   *- Do: Who talked about the convict life in Brasil in his song Care About US?*
   *- Don't: Who talked about the convict life in Brasil in his song?*

  *5. **Capture temporal information**: Ensure dates and temporal relationships are connected to relevant entities.*
   *- Attribute lifting via containment: If entity A is specified as part of/from/issued on/by entity B and B carries an explicit attribute in the same passage (e.g., date/time, location, number/ordinal), also attach that attribute to A via an appropriate relation, using the same granularity and only when unambiguous. Do not fabricate precision or lift conflicting attributes.*

  *6. **Include biographical details**: Birth/death dates, family relationships, professions, nationalities as entities.*

  *7. **Include work attributes**: For films, books, etc. capture directors, release dates, genres, etc.*

  *8. **Two questions per relationship phrase but phrases can be repeated for different entities**:*
   *- Input: "John Smith and Jane Doe were born in New York on 1990 and died in Los Angeles on 2020."*
   *- Do: "born in" and "died in" for John Smith and Jane Doe separately and 2 questions for each phrase*
   *- Don't: "born in" and "died in" for John Smith and Jane Doe together and more than 2 questions for each phrase*

  *9. **Ensure entity coverage in QAs**: Every entity node must participate in at least one verb phrase question set — either appearing in the question text or as an answer ID. If an entity would otherwise be orphaned (only present via roles/states), add a minimal factual relation to connect it (e.g., released in [date], located in [place], part of/member of [container], has role/type [concept], known as [alias]). Use only attributes stated in the passage; do not fabricate.*

*Now extract the factual relationships from the given input text STRICTLY following this pattern:*

   ***(Output format omitted here for space; see source prompt in code.)***

***Guidelines for Roles and States:***
*- **Roles**: General entity types (person, location, date, film, title, etc.)*
*- **States**: Simple status indicators (deceased, historical figure, release year, etc.)*
*- Keep roles/states general since detailed facts are captured in verb phrase questions*

*Figure 4.* Prompt used for factual GSW construction from documents (continued).

**User Prompt:**

*Given the following document and its existing GSW (Generative Semantic Workspace) structure, review and refine the GSW to fix any errors, add missing entities and verb phrase, and improve the quality of questions.*

*Refinement rules (apply all):*
*1) Preserve what's already correct.*
  *- Keep correct entities, verb phrases, and questions as-is; modify or remove incorrect items and add missing ones based strictly on the document.*
*2) Check existing entities and add new.*
  *- Ensure all entities in the document are present, including people, places, dates, titles/works, nationalities, professions, organizations, concepts, numbers/ordinals, etc.*
  *- If something that can be an entity appears only inside another entity\u2019s roles/states, or some entity is in the original document but completely missing in the existing gsw, create a separate entity node for it.*
*3) Check existing verb phrases for bidirectional QA correctness.*
  *- Every verb phrase must have two questions that are bidirectional: e.g., \"Who was born in X?\" AND \"Where was Y born?\"; \"Who directed X?\" AND \"What did Y direct?\".*
  *- QA inversion: If the A\u2192B question contains entity A and its answers are the IDs of entity B, then the B\u2192A question must contain entity B and its answers must be the IDs of entity A. The two questions must swap sides; do not repeat the same side in both questions\u2019 texts or answers.*
  *- Final QA normalization scan: correct any ill\u2011formed QAs that do not meet the above requirement. Ensure both the question text and the answer IDs correctly swap sides.*
*4) Add new verb phrases as needed to capture all relational facts stated in the document.*
*5) Ensure entity coverage in answers (final scan).*
  *- Hard requirement: Perform a single final coverage scan. Every entity ID must appear at least once in verb_phrase_nodes[*].questions[*].answers. If any entity is missing, add a minimal factual verb relation grounded in the document to include it, then finish.*
*Output the refined GSW in the same JSON format as the input, with improvements applied.*

**Inputs:**
*Input text: Original Document*
*Existing GSW Json。*

*Figure 5.* LLM prompt for 2nd pass GSW Construction.

***USER PROMPT:***
*Your task is to break down a complex multi-hop question into the most efficient sequence of single-hop, \*\*atomic\*\* questions.*
*## Your Main Goal: Build Smart Bridges, Don't Just Collect Nouns*
*The most critical skill is to convert complex logical clauses (like "despite," "the country where," "the year before") into a single, powerful \*\*bridging question\*\*. This question should use a known entity as context to find the next one. Avoid finding all the entities separately and then trying to figure out how they connect.*
*---*
*## A Simple Analogy for Efficiency*

*\*\*Question:\*\* "What is the phone number of the mother of the tallest player on the Lakers?"*

*\*\* Inefficient Path:\*\**
*1. Who are the players on the Lakers?*
*2. What are all their heights?*
*3. Who is the mother of the tallest player?*

*\*\* Efficient Path:\*\**
*1. Who is the tallest player on the Lakers?*
*2. Who is the mother of `<ENTITY_Q1>`?*
*3. What is the phone number of `<ENTITY_Q2>`?*
*---*
*## How to Decompose a Question*

*### 1. Analyze the Query's Components*
*First, break down the original question into its fundamental building blocks. Identify the core \*\*entities\*\* (people, places, organizations), their \*\*properties\*\* (attributes like rank, location, date), and the \*\*relationships\*\* that connect them.*

*### 2. Construct an Atomic Chain*
*Next, formulate a sequence of questions where each question retrieves a single fact.*
*- \*\*Isolate Comparisons:\*\* Don't ask "who is faster?" Ask for the specific rank or time.*
*- \*\*Link with Placeholders:\*\* Use `<ENTITY_Qn>` to pass the answer from a previous question (`Qn`) into the next one.*

*### 3. Optimize for Efficiency and Precision*
*Your final goal is the \*\*shortest and most direct path\*\* to the answer.*
*- \*\*Embed Constraints to Build Bridges:\*\* If a piece of information is only a filter (date, location), embed it as a constraint in the next question instead of asking for it directly.*
*- \*\*Important note for bridges:\*\* There can be no `<ENTITY_Qn>` in the first question if the nth question DOES NOT require retrieval.*

*## Formatting*
*Format each decomposed question as follows:*

*Question: [the question text]*
*Requires retrieval: [True/False]*

*And provide the response in the following JSON format:*
*{*
*  "questions": [*
*    {*
*      "question": "the decomposed question text",*
*      "requires_retrieval": "True/False"*
*    }*
*  ]*
*}*
***(Examples omitted here for space; see source prompt in code.)***
*IMPORTANT:*
*  AVOID over-decomposition like this:*
*  DON'T break "Who is John Doe?" into:*
*  1. Who is John Doe? → "English"*
*  2. When was <ENTITY_Q1> born? → "When was English born?"*

*  DO ask directly: "When was John Doe born?"*

*Now decompose this question:*
*Input: "{QUESTION}"*
*Output:*

*Figure 6.* Prompt used for multi-hop question decomposition into atomic sub-questions.

**SYSTEM PROMPT:**
*As an advanced reading comprehension assistant, your task is to analyze precise QA pairs extracted from the documents and corresponding questions meticulously.*
*Your response start after "Thought: ", where you will methodically break down the reasoning process, illustrating how you arrive at conclusions.*
*Conclude with "Answer: " to present only a concise, definitive response, devoid of additional elaborations.*
*(If platinum and unanswerable datasets getting evaluated: If you don't know the answer, say "No Answer")*

**USER PROMPT:**
*Q: Who directed The Last Horse? A: Edgar Neville*
*Q: When was The Last Horse released? A: 1950*
*Q: When was the University of Southampton founded? A: 1862*
*Q: Where is the University of Southampton located? A: Southampton*
*Q: What is the population of Stanton Township? A: 505*
*Q: Where is Stanton Township? A: Champaign County, Illinois*
*Q: Who is Neville A. Stanton? A: British Professor of Human Factors and Ergonomics*
*Q: Where does Neville A. Stanton work? A: University of Southampton*
*Q: What is Neville A. Stanton's profession? A: Professor*
*Q: Who directed Finding Nemo? A: Andrew Stanton*
*Q: When was Finding Nemo released? A: 2003*
*Q: What company produced Finding Nemo? A: Pixar Animation Studios*

*Question: When was Neville A. Stanton's employer founded?*
*Thought:*

**ASSISTANT:**
*From the QA pairs, the employer of Neville A. Stanton is University of Southampton. The University of Southampton was founded in 1862.*
*Answer: 1862.*

**USER:**
*{EVIDENCE_QA_PAIRS}*
*Question: {QUESTION}*

*Thought:*

*Figure 7.* Oracle-style prompt used for answer generation from retrieved GSW evidence.

---

**End-to-End Retrieval Trace (Step 1/3): Question Decomposition**

---

**Question.** What is the place of birth of the performer of song *Changed It*?
**System.** Chain-Following Multi-Hop QA with Beam Search    **Mode.** cumulative ($\alpha = 0.5$), reranker = voyage    **beam_width** $= 5$, chain_top_k $= 15$, entity_top_k $= 20$, qa_rerank_top_k $= 15$

**Original question.** "What is the place of birth of the performer of song Changed It?"
**Decomposed sub-questions.** (1) **Q1:** "Who performed the song Changed It?" [Retrieval: ✓] → (2) **Q2:** "What is the place of birth of `<ENTITY_Q1>`?" [Retrieval: ✓]
**Dependencies.** Q1 has no dependencies; Q2 depends on Q1.
**Chain identification.** Total chains = 1; Chain 1 indices = [0,1]; Questions = Q1 → Q2.

---

**End-to-End Retrieval Trace (Step 2.1/3): Retrieval Hop 1 with Beam Search**

---

**Hop 1/2 (Q1):** "Who performed the song Changed It?"
**State.** Prior entities = None (root); Active beams = 1; Concrete questions = 1.

**Top retrieved entities (BM25 + Embedding; top-5 of 20):**

| Rank | Entity name |
|---|---|
| 1 | Changed It |
| 2 | Liar, Liar |
| 3 | The Supremes |
| 4 | You Changed Me |
| 5 | When the Stars Go Blue |

**Retrieved QA pairs after reranking (top-5 of 15; remaining omitted):**

| Rank | Question | Answer | Source |
|---|---|---|---|
| 1 | Who performed "Changed It"? | Nicki Minaj, Lil Wayne | Changed It |
| 2 | What song did Nicki Minaj and Lil Wayne perform? | Changed It | Changed It |
| 3 | What song did Motiv, Detail, and Sidney Swift produce? | Changed It | Changed It |
| 4 | What song did Young Money Entertainment, Cash Money Records, Republic Records release? | Changed It | Changed It |
| 5 | What song was released on March 10, 2017? | Changed It | Changed It |

**Beam expansion.** Group by answer entity and keep the best QA pair per entity; unique answer entities = 17.
**Sample candidate states (first 5):**

| Cand. | last_hop | Answer entity/entities |
|---|---|---|
| 1 | 0.8320 | [Nicki Minaj, Lil Wayne] |
| 2 | 0.7852 | [Changed It] |
| 3 | 0.6953 | [Changed It] |
| 4 | 0.6875 | [Changed It] |
| 5 | 0.6172 | [Changed It] |

**Beam pruning (Hop 1).** Total candidates = 17; keep top-5 by (chain_score, last_hop_score).
**Beam details (Hop 1).** Includes all kept beams plus the top-ranked pruned beam.

| Beam | chain_score | last_hop | Status | Entity map |
|---|---|---|---|---|
| 1 | 0.9160 | 0.8320 | ✓ kept | {0: Nicki Minaj} |
| 2 | 0.9160 | 0.8320 | ✓ kept | {0: Lil Wayne} |
| 3 | 0.8926 | 0.7852 | ✓ kept | {0: Changed It} |
| 4 | 0.8086 | 0.6172 | ✓ kept | {0: No Frauds} |
| 5 | 0.8086 | 0.6172 | ✓ kept | {0: Regret in Your Tears} |
| 6 | 0.7949 | 0.5898 | ✗ pruned | {0: March 10, 2017} |

---

**End-to-End Retrieval Trace (Step 2.2/3): Retrieval Hop 2 with Beam Search**

**Hop 2/2 (Q2):** "What is the place of birth of `<ENTITY_Q1>`?"
**State.** Prior entities = {0: Nicki Minaj}; Active beams = 5; Concrete questions = 5 (Nicki Minaj; Lil Wayne; Changed It; No Frauds; Regret in Your Tears).
**Top retrieved entities (BM25 + Embedding; top-5 of 20):**

| Rank | Entity name |
|---|---|
| 1 | Nicki Minaj |
| 2 | Nicki Minaj |
| 3 | Pink Friday |
| 4 | Nicki Minaj |
| 5 | Nicki Minaj |

**Retrieved QA pairs after reranking (top-5 of 15; remaining omitted):**

| Rank | Question | Answer | Source |
|---|---|---|---|
| 1 | Where was Nicki Minaj born? | Saint James, Port of Spain | Nicki Minaj |
| 2 | Who was born in Saint James, Port of Spain? | Nicki Minaj | Nicki Minaj |
| 3 | Who was raised in Queens, New York City? | Nicki Minaj | Nicki Minaj |
| 4 | Where was Nicki Minaj raised? | Queens, New York City | Nicki Minaj |
| 5 | Who performed "Changed It"? | Nicki Minaj, Lil Wayne | Nicki Minaj |

**Beam expansion (final hop).** Use all QA pairs directly (no grouping); candidates = 15.
**Sample candidate states (first 5):**

| Cand. | last_hop | Answer entity/entities |
|---|---|---|
| 1 | 0.8945 | [Saint James, Port of Spain] |
| 2 | 0.7891 | [Nicki Minaj] |
| 3 | 0.6836 | [Nicki Minaj] |
| 4 | 0.6289 | [Queens, New York City] |
| 5 | 0.5898 | [Nicki Minaj, Lil Wayne] |

**Beam pruning (Hop 2).** Total candidates = 15; keep top-5 by (chain_score, last_hop_score).
**Beam details (Hop 2).** Includes all kept beams plus the top-ranked pruned beam.

| Beam | chain_score | last_hop | Status | Entity map |
|---|---|---|---|---|
| 1 | 0.9315 | 0.8945 | ✓ kept | {0: Nicki Minaj, 1: [Saint James, Port of Spain]} |
| 2 | 0.9052 | 0.7891 | ✓ kept | {0: Nicki Minaj, 1: [Nicki Minaj]} |
| 3 | 0.8781 | 0.6836 | ✓ kept | {0: Nicki Minaj, 1: [Nicki Minaj]} |
| 4 | 0.8637 | 0.6289 | ✓ kept | {0: Nicki Minaj, 1: [Queens, New York City]} |
| 5 | 0.8533 | 0.5898 | ✓ kept | {0: Nicki Minaj, 1: [Nicki Minaj, Lil Wayne]} |
| 6 | 0.8406 | 0.5430 | ✗ pruned | {0: Nicki Minaj, 1: [Onika Tanya Maraj-Petty]} |

---

**End-to-End Retrieval Trace (Step 3/3): Answer Generation from Final Evidence**

**Final evidence (5 unique QA pairs).**
(1) Q: Who performed "Changed It"?   A: Nicki Minaj, Lil Wayne   person: Trinidadian-American, rapper, singer; person: American, rapper
(2) Q: Where was Nicki Minaj born?   A: Saint James, Port of Spain   location: birthplace, Trinidad and Tobago
(3) Q: Who was born in Saint James, Port of Spain?   A: Nicki Minaj   person: stage name, musician, actress, model
(4) Q: Who was raised in Queens, New York City?   A: Nicki Minaj   person: stage name, musician, actress, model
(5) Q: Where was Nicki Minaj raised?   A: Queens, New York City   location: raised in, USA

**LLM response.**
*"The performer of the song 'Changed It' is Nicki Minaj. According to the QA pairs, Nicki Minaj was born in Saint James, Port of Spain, which is located in Trinidad and Tobago. Therefore, the place of birth of the performer is Saint James, Port of Spain.*
*Answer: Saint James, Port of Spain."*

---

*Figure 8.* **Example end-to-end retrieval trace.** PANINI decomposes the query into two hops and performs chain-following retrieval with beam search (beam width = 5). At each hop, candidates are scored using geometric-mean (cumulative) chain scoring, and the final answer is generated from the selected QA evidence.

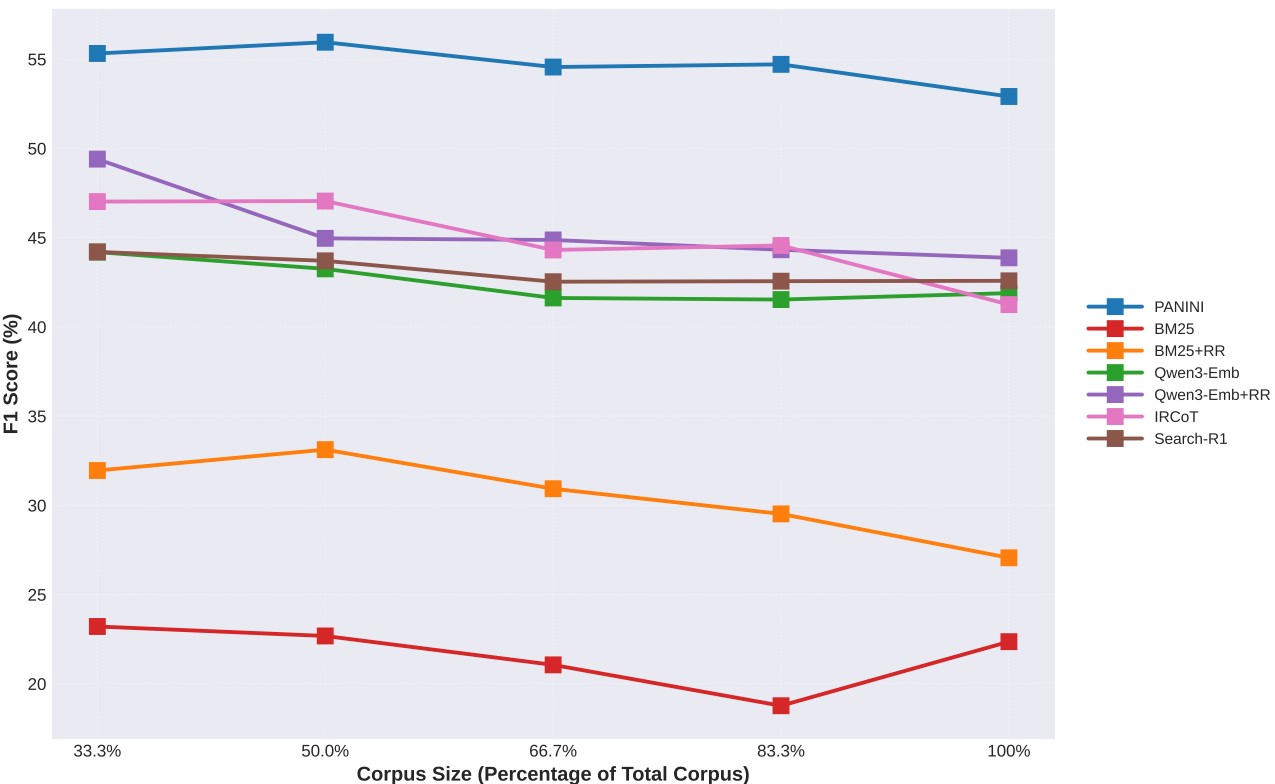

*Figure 9.* **Continual-learning under corpus growth (MuSiQue, 200 questions).** We fix a held-out set of 200 questions and evaluate retrieval+QA F1 as the corpus is incrementally expanded from 4K to the full MuSiQue collection (∼12K passages). The set of *relevant* passages for these questions is contained in the initial 4K subset and remains unchanged across steps; only the number of *distractor* passages increases as additional (irrelevant) documents are added. This simulates a continuously evolving corpus where the signal stays constant but the retrieval search space grows.

*Table 10.* Performance comparison when varying the open-source model used for write-time GSW construction. All open-source models use the same read-time configuration: LoRA-finetuned Qwen3-8B for question decomposition and Qwen3-8B (no-think) as the reader. We also report HippoRAG2 under the same reader model and the PANINI configuration from Table 2 as reference points.

| Model | 2Wiki | MuSiQue |
|---|---|---|
| Qwen3-8B | 55.38 | 40.24 |
| Qwen3-14B | 62.80 | 44.87 |
| Qwen3-14B (Second Pass) | 66.76 | 48.28 |
| GPT-OSS-120B | 70.65 | 48.50 |
| HippoRAG2 (Qwen3-8B reader) | 66.93 | 45.40 |
| PANINI (Main Exp) | 72.4 | 52.3 |

*Table 11.* Ablation study on multi-hop QA (F1 score, 1000 questions).

| Setting | MuSiQue | 2Wiki |
|---|---|---|
| **Full PANINI** | **52.3** | **72.4** |
| No QA reranking | 18.4 | 22.2 |
| No dual search (entity-only) | 39.5 | 68.6 |
| No dual search (QA-only) | 52.3 | 72.0 |
| No decomposition | 36.8 | 47.2 |
| Beam width = 10 | 52.4 | 72.7 |
| Beam width = 3 | 52.3 | 72.2 |
| Beam width = 1 | 44.6 | 67.3 |

*Table 14.* Controlled MuSiQue retrieval ablation over evidence interfaces. Direct dense retrieval over the same GSW QA pairs performs worse than RICR, indicating that the gain is not explained by QA evidence formatting alone.

| Method | F1 ↑ |
|---|---|
| Raw document chunks + vector retrieval + reranker | 43.7 |
| GSW QA pairs + vector retrieval + reranker | 34.2 |
| PANINI (RICR over GSW QA pairs) | **52.3** |

*Table 12.* Ablation study on multi-hop QA (F1 score and average token count, 1000 questions).

| Setting | MuSiQue | | 2Wiki | |
|---|---|---|---|---|
| | F1 | Tokens | F1 | Tokens |
| **Beam width = 5** | 52.3 | 192 | 72.4 | 315 |
| Beam width = 10 | **52.4** | 320 | **72.7** | 409 |
| Beam width = 3 | 52.3 | 143 | 72.2 | 231 |
| Beam width = 1 | 44.6 | 82 | 67.3 | 171 |

*Table 13.* **Ablation of chain-level scoring on MuSiQue** (**F1**, 1,000 questions). "Main" denotes cumulative (geometric-mean) scoring; "Similarity" uses only question–chain embedding similarity; "Combined" linearly interpolates the two ($\alpha = 0.5$); "None" disables chain-level scoring and performs greedy last-hop selection.

| Scoring rule | MuSiQue |
|---|---|
| None (greedy last-hop) | 50.81 |
| Similarity | 42.32 |
| Combined | 48.24 |
| **Main (cumulative)** | **52.3** |

*Table 15.* Direct GPT-4o-mini answer baseline without retrieved evidence.

| Method | NQ | PopQA | MuSiQue | 2Wiki | HotpotQA | LV-Eval | Avg |
|---|---|---|---|---|---|---|---|
| No retrieval | 52.7 | 22.7 | 22.0 | 36.3 | 41.0 | 5.0 | 29.9 |
| PANINI | **67.4** | **57.6** | **52.3** | **72.4** | **71.9** | **14.8** | **56.1** |

*Table 16.* **Write-time indexing cost on MuSiQue (11,656 passages).** Costs are one-time dataset-level indexing costs. PANINI write-time corresponds to per-document GSW construction with `gpt-4.1-mini`; the Batch variant uses the OpenAI Batch API, while the Standard variant is an estimate assuming a 2× higher cost without the Batch discount. HippoRAG 2 cost is the reported total run cost using `GPT-4o-mini` with standard API calls as default. Qwen3-Embedding (8B) has zero write-time LLM cost since it performs embedding-only indexing.

| Method | Write-time cost |
|---|---|
| PANINI (Batch) | **$48.02** |
| PANINI (Standard, est.) | $96.04 |
| HippoRAG 2 | $91.8 |
| Qwen3-Embedding (8B) | $0 |

*Table 17.* Performance of **Search-R1** under different retrieval configurations. All rows use the same Search-R1 agent model and differ only in the retrieval mechanism. **Search-R1** uses BM25 over document chunks; **Search-R1 (Dense)** replaces BM25 with dense embedding retrieval (`Qwen3-Embedding-8B`) over document chunks; **Search-R1 (Dense + QA)** further replaces document chunks with the structured QA pairs from the GSW; **Search-R1 (Dense + QA + Dual Index)** additionally incorporates PANINI's RICR retrieval strategy and retrieves relevant QA pairs. **PANINI** is included as a reference point and uses the same evaluation setting as in Table 2.

| | Simple QA | | Multi-Hop QA | | | | Avg |
|---|---|---|---|---|---|---|---|
| Retrieval | NQ | PopQA | MuSiQue | 2Wiki | HotpotQA | LV-Eval | Avg |
| Search-R1 | 47.9 | 49.7 | 41.1 | 64.9 | 68.6 | 11.5 | 47.3 |
| Search-R1 (Dense) | 41.6 | 52.7 | 45.1 | 64.6 | 71.3 | 18.8 | 49.0 |
| Search-R1 (Dense + QA) | 48.5 | 51.5 | 44.6 | 65.4 | 66.8 | 12.0 | 48.1 |
| Search-R1 (Dense + QA + Retrieval) | 48.1 | 54.0 | 46.6 | 67.6 | 70.0 | 10.5 | 49.4 |
| PANINI | 67.4 | 57.6 | 52.3 | 72.4 | 71.9 | 14.8 | 56.1 |

*Table 18.* Representative failure cases for dense embedding retrieval (Qwen3-Embedding-8B) and graph-based retrieval (HippoRAG2).

| Query | Method | Failure mode | Failure reason |
|---|---|---|---|
| What year did the publisher of *Labyrinth* end? | Qwen3-Emb | `missing_second_hop` | Retrieved passages broadly about *Labyrinth* and publishing, but missed the bridging evidence required for the second hop. |
| What is the location of formation of the film company distributing *The Boss*? | Qwen3-Emb | `missing_second_hop` | Over-matched the surface phrase "The Boss" and returned loosely related documents, failing to retrieve the distributor → company-formation link. |
| Between the state university in the state without North Point Mall and where Edwards won the primary and the university in Fort Hill's town, which has the more national championships? | Qwen3-Emb | `missing_entity` | Requires multiple supporting documents; failed to retrieve the intermediate evidence involving North Point Mall, breaking the chain. |
| Who's the son of the Italian navigator who explored the eastern coast of the continent Manuel Balbi was born in? | Qwen3-Emb | `broken_chains` | Retrieved the correct answer passage, but missed the intermediate evidence (continent identification and the relevant navigator), so the chain could not be supported. |
| What is the name of the famous bridge in the birth city of the composer of *Scanderbeg*? | HippoRAG2 | `retrieval_drift` | Drifted to a plausible distractor in the same city (e.g., another famous Venice bridge), and failed to rank the correct bridge-related evidence highest. |
| Who is the child of the president under whom prohibition occurred? | HippoRAG2 | `retrieval_drift` | Found evidence about prohibition but drifted to nearby presidents/laws, losing the correct intermediate link needed to answer the composed query. |
| What is the highest city in the state where Dell ranks sixth by revenue? | HippoRAG2 | `missing_second_hop` | Stayed in Dell-centric neighborhoods and failed to hop from company revenue ranking → state → highest city evidence. |

*Table 19.* Representative qualitative error cases for PANINI.

| Failure mode | Example query | What goes wrong | Why it breaks the chain |
|---|---|---|---|
| Missing verb-phrase nodes | What nationality is Arabia (Daughter Of Justin II)'s mother? | The GSW omits the verb-phrase/event capturing *Aelia Sophia* as an *empress of the Byzantine Empire*. | Without this relation node, the GSW does not contain any QA pair linking the mother to a country or polity, so the required evidence for determining nationality is never retrieved. |
| QA-pair construction errors | Who is the paternal grandfather of Dibyasambandh? | The relation *daughter of* is stored, but QA pairs are only generated in the father→daughter direction; the inverse daughter→father QA is missing. | Since the father entity never appears as the answer to any QA pair, subsequent hops remain conditioned on the daughter entity, blocking retrieval of evidence about the paternal grandfather. |
| Imperfect question decomposition | When did the country that has the original language of the film named after Vladimir Karalić's birthplace as a *co-official* language first attend the Olympics as an independent team? | The decomposer drops *co-* and queries for *official* language instead of *co-official* language. | The weakened constraint retrieves a different country, causing subsequent hops to follow an incorrect chain even if later retrieval is accurate. |

*Table 20.* **Computational resource requirements on MuSiQue (11,656 passages).** We report indexing time and QA time per query for RAG baselines. Percentages are relative to PANINI (100%). **Note:** Results for NV-Embed-v2, GraphRAG are from HippoRAG 2 (Gutiérrez et al., 2025b).

| | NV-Embed* | GraphRAG* | HippoRAG 2 | Search-R1 | PANINI |
|---|---|---|---|---|---|
| **Index (min)** | 12.1 | 277.0 | 106.6 | 12.1 | **100.1** |
| **QA (sec)** | 0.3 | 10.7 | 4.4 | 6.72 | **3.3** |

