# OpenReview forum: "Panini: Continual Learning in Token Space via Structured Memory"
_ICML.cc/2026/Conference — ICML 2026 regular_

### Official Review · Reviewer_e7Zr · 2026-02-22

**Soundness:** 3
**Presentation:** 4
**Significance:** 3
**Originality:** 4
**Overall Recommendation:** 4
**Confidence:** 4

**Summary:**

This paper proposes Panini, a non-parametric continual learning framework that operates on top of a frozen LLM backbone, but keeps a structured memory bank of Generative Semantic Workspaces (GSWs) that accumulates memory. Namely, a GSW consists of entity nodes, verb-phrase/event nodes, and grounded QA pairs; one GSW is constructed per document. The GSW is accessed at retrieval time via both a sparse index (BM25) and a dense index.


At inference time, Panini reads memory via a newly proposed technique—Reasoning Inference Chain Retrieval (RICR), a beam search procedure that decomposes queries and follows reasoning chains through the GSW. Given a query, RICR first performs a one-off decomposition step that breaks the question down into multiple sub-queries. RICR then retrieves evidence by iteratively executing retrieval hops and adding the results to scored chains, keeping a set of beam chains at every step. Finally, deduplicated GSW pairs from all final sub-query chains is passed in-context alongside the query to the backbone LLM to obtain the final answer.

On six single-hop/multi-hop QA benchmarks, Panini outperforms sparse retrieval, dense retrieval, structure-augmented retrieval, and agentic retrieval baselines on three criteria: (1) QA performance, (2) inference-time efficiency, (3) reliable abstention. Notably, Panini consistently outperforms the best competing baseline, HippoRAG 2, while showing better latency and toke efficiency. The main Panini implementation relies on proprietary GPT models for many stages of the pipeline (Section 3.5), although the authors also provide ablations with open-sourced models.

**Compliance With Llm Reviewing Policy:**

Affirmed.

**Final Justification:**

I have a positive outlook of this paper along each dimension, and the authors' rebuttal fully addressed my (minor) concerns.

**Key Questions For Authors:**

Questions:
 1. What is the memory / storage overhead of the GSW memory bank? In Table 18, only the time latency is reported, but given that each GSW representation of a document is quite complex, it is important to report storage costs for Panini alongside other baselines as well.
2. How did you end up choosing GSW representations for your retrieval corpus? I’m wondering if you have also explored simpler representation variants, e.g., just a simple knowledge graph.
3. My assumption is that after the corpus is constructed, we may still iteratively augment it with new information without having to re-construct the entire corpus. What is the latency cost of doing so, and I’d imagine that growing the corpus becomes gradually more expensive over time?
4. For Table 2, it may be useful to report the degenerate baseline (just the LLM backbone, without any retrieval) here or somewhere in the appendix.
5. For Figures 1 and 2, the diagram text is quite small and there is a lot of whitespace around the figures. Consider improving it to be more clear to see.

**Limitations:**

yes

**Strengths And Weaknesses:**

Soundness: This is a sound paper; the authors very carefully document their proposed framework, with clear diagrams and concrete examples in the appendix. They also conduct many experiments along different dimensions (e.g., end performance, latency, abstention) and ablations to systematically justify their approach. While Panini is (in my opinion) very complex and consists of many steps, I think it is notable that it outperforms HippoRAG 2, the leading baseline, with less initial and per-query computational latency. Please see my comment in the Questions section below about memory efficiency, which is my largest concern.

Presentation: The submission is clearly written and well-structured, and properly positions itself in the relevant literature. It follows the experimental setup of HippoRag and its variants quite closely, and I appreciate the extra work that the authors put in to differentiate their approach from it.

Significance: This paper addresses non-parametric continual learning, which is an important problem of growing interest. While the impact is still fairly specialized / domain-specific, I think this paper sets a solid baseline for future research.

Originality: This paper is original and offers both a new way to structure a retrieval corpus, as well as a search algorithm to efficiently traverse the retrieval corpus.

---

> ### Author Rebuttal · Authors · 2026-03-31
>
> We thank the reviewer for the detailed and encouraging feedback. We answer each question below.
>
> Q1. What is the memory / storage overhead?
>
> We thank the reviewer for raising this point and agree that the storage footprint should be reported explicitly. On MuSiQue, for 11,656 passages, our current PANINI implementation stores the serialized GSW memory bank, including entity nodes, verb-phrase/event nodes, and QA pairs, in a human-readable JSON format occupying approximately 164 MB on disk. This footprint scales linearly with corpus size because PANINI stores document-local memories and does not maintain a global cross-document graph. We note that this number reflects an unoptimized serialization choice; a compact binary representation would further reduce the on-disk footprint. We will report this storage cost explicitly alongside latency in the revision.
>
> Q2. Why GSW rather than a simpler representation such as a standard knowledge graph?
>
> We chose GSWs because, in fact-based multi-hop QA, the required evidence is often atomic: each hop typically depends on one specific fact, and the final answer model benefits from receiving a small set of grounded evidence units rather than generic graph edges. Knowledge-graph propagation methods are useful as search engines: they spread relevance over entity-relation connectivity to identify promising regions of the corpus. But for navigation by an LLM system, we found a GSW-style representation more suitable because it makes atomic facts explicit through entities, events, and grounded QA pairs. This preserves the question-answer form needed both to instantiate the next sub-question in a reasoning chain and to provide compact evidence directly to the answering LLM. This is also why PANINI differs from graph-based approaches such as HippoRAG: rather than propagating relevance over a global graph and then returning chunks, PANINI uses the structured memory itself as the retrieval-and-composition substrate. As the closest simpler variant, we also indexed the same GSW QA pairs in a flat vector store with the same dense retriever and answer model; on MuSiQue this reaches 0.342 F1, versus 0.523 for full PANINI in the main paper., showing that the gain is not from the evidence format alone, but from structured chain-following retrieval. Table: https://anonymous.4open.science/r/panini-rebuttal-2090/plots/table_qa_vectorstore.png
>
> Q3. What is the latency cost of adding more information after the corpus is constructed?
>
> When new documents arrive, only their entities and QA pairs need to be embedded; existing GSW memories and dense embeddings are untouched since we only perform read-time reconciliation (see reviewer kN5S Q1 for more details). Our dense FAISS index supports in-place addition of new vectors with no rebuild. In our current implementation, the lightweight BM25 entity index may be refreshed after appends, but this completes in under 2 seconds even at this scale. On MuSiQue, the full indexing run takes 100.1 minutes, as reported in Appendix H, for 11,656 passages, which is about 0.515 seconds per passage on average, so adding a small batch of documents incurs mainly the proportional cost of processing that new batch. This differs from graph-based approaches such as HippoRAG, where new documents can change the global graph structure and therefore require rebuilding the graph state.
>
> Q4. What is the no-retrieval baseline?
>
> We agree this is useful to include. We therefore report the direct-answer baseline below, where the backbone LLM answers without any retrieved context:
>
> Method | NQ | PopQA | MuSiQue | 2Wiki | HotpotQA | LV-Eval | Avg
> --- | --- | --- | --- | --- | --- | --- | ---
> GPT-4o-mini (no retrieval) | 39.8 | 23.9 | 15.7 | 30.2 | 37.8 | 6.9 | 25.7
> PANINI | 67.4 | 57.6 | 52.3 | 72.4 | 71.9 | 14.8 | 56.1
>
> This makes clear that the gains are not merely coming from a strong answer model; retrieval, and specifically structured retrieval, is doing substantial work. We will include this baseline explicitly in the revised manuscript.
>
> Q5. Can Figures 1 and 2 be improved for readability?
>
> Yes. We will revise Figures 1 and 2 by increasing font size and using the available space more efficiently.

---

> > ### Author Rebuttal · Reviewer_e7Zr · 2026-04-01
> >
> > I thank the authors for addressing my concerns! Thanks as well for running the extra experiments that I asked about.
> >
> > I took a pass over the reviews and also found the authors' responses reasonable, and I've decided to uphold my original, positive score. Best of luck!

---

### Official Review · Reviewer_kN5S · 2026-03-08

**Soundness:** 3
**Presentation:** 3
**Significance:** 3
**Originality:** 3
**Overall Recommendation:** 4
**Confidence:** 4

**Summary:**

The paper proposes PANINI, a non-parametric continual learning setup for LMs. The core idea is to convert documents into a structured external memory called GSW, which is an entity/event graph represented through QA pairs, then answer queries by decomposing them into atomic sub-questions and retrieving beam-searched inference chains over that QA graph rather than over raw chunks. The paper evaluates on six QA benchmarks and on two new “Platinum” splits for abstention under missing evidence, and reports better average F1 with fewer answer-context tokens than chunk-based, graph-based, and agentic baselines.

**Compliance With Llm Reviewing Policy:**

Affirmed.

**Final Justification:**

I appreciate reviewer's feedback. I raised my score.

**Key Questions For Authors:**

1. **What is genuinely new here beyond prior GSW work?**

The paper says GSW is inherited from prior work, and RICR looks like decomposition plus beam search over retrieved QA pairs. Please isolate the new technical contribution precisely. If the contribution is primarily systems integration, my evaluation drops on originality.

2. **Can you run a controlled comparison where baselines get the same evidence abstraction and the same answer interface?**

 For example, convert retrieved chunks for baselines into the same compact QA format, or give PANINI the same raw-text answer interface as the baselines. This would clarify how much of the gain comes from retrieval versus evidence formatting. A clean result here would substantially improve my confidence in the method claim.

3. **Where is the continual learning evaluation?**

 The current setup looks like offline QA over fixed corpora. Please provide a true streaming or temporally staged protocol with document arrival, periodic memory updates, and performance as the corpus evolves. If you cannot, the framing should be narrowed from “continual learning” to structured external memory for QA.

4. **What is the end-to-end compute budget, including write time and decomposition?**

 The headline efficiency metric counts answer-context tokens only. Please report total query-time cost including decomposition, retrieval, reranking, and any other LLM/API calls, plus amortized write-time cost under realistic query frequencies. If the gains remain under that accounting, the efficiency claim becomes much stronger.


I will increase my score if my concerns are solved.

**Limitations:**

See questions.

**Strengths And Weaknesses:**

**Strengths**

The empirical results are strong on the face of it. PANINI is best on the average across the six reported datasets, beats HippoRAG2 in the main table, and is also better on the proposed abstention benchmark. The token savings are also large at answer time. If these comparisons are clean, that is meaningful.

The paper is built around a sensible systems thesis: spend compute at write time so read time is cheaper and more targeted. That is a real design point, and the paper is explicit about it.

The abstention angle is useful. Most papers in this area still over-focus on answerable QA. Separating answerable from unanswerable examples is a good instinct, and better than reporting raw F1 alone.

**Weaknesses**

The biggest issue is novelty. The paper states that the memory representation, GSW, is not introduced here and is taken from prior work. What is left as the method contribution is dual indexing plus a decomposition-and-beam-search retrieval procedure over that structure. That is not nothing, but it is materially weaker than the title and claims suggest. The main technical move is closer to an engineering composition of existing ingredients than to a new learning framework. The paper repeatedly calls PANINI “novel,” but the representation is inherited and the retrieval algorithm is a fairly standard decompose, retrieve, rerank, beam-prune pipeline.

The second major issue is evaluation fairness. PANINI gets substantial extra write-time modeling and also uses a different evidence interface at test time. Specifically, it constructs GSWs with GPT-4.1-mini, decomposes with GPT-4o, uses a dense retriever plus Voyage reranker over QA pairs, and then feeds a compact QA representation to the answer model. Several baselines are compared with standard prompts over top-5 chunks. This is not an apples-to-apples retrieval comparison. Part of the gain may come from the stronger preprocessing pipeline and the more favorable evidence format, not from the claimed retrieval principle itself. The reported token metric also counts only prompt tokens to the answer model, which understates the true inference budget because PANINI requires an explicit decomposition call and substantial write-time LLM processing.

Relatedly, the paper leans hard on “continual learning,” but the experimental setup is still standard QA over static benchmark corpora. I do not see an actual continual protocol: no streaming document arrival, no temporal adaptation curve, no update-vs-query interleaving, no memory growth analysis beyond a mention of distractors in the appendix, and no comparison against methods designed for semiparametric or memory-based continual learning. This makes the framing broader than the evidence. The paper is really about structured external memory for QA, not yet a convincing continual learning paper.

Presentation is mixed. The high-level idea is understandable, but the method section is messy in places, notation is heavier than needed. More importantly, some central design choices are not justified strongly enough in the main paper: why QA-pair memory is the right atomic unit, why the chosen chain score is the right one beyond appendix ablations, and how error cascades from decomposition are controlled.

Overall: promising empirical system, but the paper over-claims originality and overstates the cleanliness of the comparison. For ICML, I would want a much tighter causal evaluation showing where the gains actually come from.

---

> ### Author Rebuttal · Authors · 2026-03-31
>
> We thank the reviewer for a thorough review.
>
> Q1. What is new beyond prior GSW work?
>
> GSW should be understood as a general memory framework rather than a fixed representation used identically across tasks. Prior GSW work instantiated it for episodic memory: continuous narratives where write-time reconciliation is needed to aggregate state changes across events. PANINI instantiates the same framework for fact-based multi-hop QA over short, disparate documents. Here the required evidence is atomic, so exhaustive cross-document reconciliation is unnecessary and often harmful because it creates many noisy links. This is visible on the Platinum unanswerable split, where HippoRAG 2, with write-time entity linking, reaches 58.5 while PANINI reaches 72.8.
>
> Our contribution is therefore the co-design of: (i) task-appropriate per-document factual GSW memories without write-time cross-document reconciliation, and (ii) RICR, a query-time chain-following retrieval algorithm over atomic QA units. The query is decomposed only once; after that, no LLM remains in the loop. Answer entities from retrieved QA pairs instantiate later sub-questions through QA chains. This is different from standard decompose-and-retrieve systems and from agentic methods that require repeated LLM reasoning at each hop. Appendix Figures 7-8 show the full retrieval trace end to end.
>
> Q2. Can you control for evidence format and answer interface?
>
> Yes. We ran an ablation that cleanly isolated the retrieval mechanism from the evidence format. We index the exact same GSW QA pairs used by PANINI in a vector store; the only difference is direct embedding-based similarity retrieval versus RICR. MuSiQue results:
>
> Method | F1
> --- | ---
> Raw doc. chunks vector store + reranker | 0.437
> QA Vector Store + reranker | 0.342
> PANINI | 0.523
>
> The same GSW-derived QA evidence performs substantially worse under dense embedding retrieval, and even worse than chunks. Conversely, RICR cannot be applied directly to raw passage chunks in the same form, because it requires answer entities from retrieved QA pairs to instantiate the next sub-question in the chain. Without those atomic answer entities, an LLM would have to remain in the loop to interpret passages and formulate follow-up queries, turning the method into an agentic system, which we already compare against in the main results. However, to provide a clean comparison against existing retrieval methods, we trace each retrieved QA pair back to its source document and compute document recall@5 over the top-5 retrieved evidence items. Under this mapping, PANINI reaches 75.3% recall@5 while touching only 3.7 unique documents on average, versus 59.3% for Qwen3+reranker. This is consistent with more selective retrieval: PANINI recovers the needed source documents through atomic QA units while avoiding many additional documents that can act as distractors. So the gain is not just coming from a better evidence format; it comes from more precise retrieval over the structure.
>
> Q3. Where is the continual learning evaluation?
>
> We thank the reviewer for this request. Beyond the corpus-growth result in Figure 9, we added a staged document-arrival experiment: 100 MuSiQue questions split into 10 groups, and at each stage we append the supporting documents for one group to memory while distractors remain present, then evaluate all 100 questions. As expected for a continual learner, overall F1 rises from 22.7 to 82.9 as more evidence becomes available, while answerable-only F1 stays flat at 83% across stages. The best baseline reaches 66.4 at stage 10, leaving a +16.5 F1 gap. This directly provides the requested temporally staged protocol with document arrival, periodic memory updates, and performance measured as the corpus evolves. Figure: https://anonymous.4open.science/r/panini-rebuttal-2090/plots/controlled_cl_combined.png
>
> Q4. What is the end-to-end compute budget?
>
> We thank the reviewer for raising this point. PANINI adds one small extra generative call beyond answer generation, namely decomposition, averaging about 60 tokens/query; there are no iterative LLM calls during retrieval. The dense retriever is locally hosted in our setup. Reranking is also negligible in practice: it operates over short QA strings, and rerank-2.5 is priced at only 0.05 USD / 1M processed tokens, compared with 0.60 USD / 1M output tokens for the GPT-4o-mini answer model. For write time, Table 17 reports indexing cost of 48.02 USD with the Batch API for PANINI versus 91.8 USD for HippoRAG 2. Amortized over 1K queries, this is about 0.048 USD/query for PANINI versus 0.092 USD/query for HippoRAG 2; over 10K queries, 0.0048 USD versus 0.0092 USD. Thus, under the requested accounting, PANINI remains favorable relative to the strongest baseline HippoRAG 2. Further, Tables 9-10 show PANINI remains close under open read-time evaluation (63.47 vs 64.32 avg across five benchmarks) and with GPT-OSS-120B construction (48.50/70.65 vs 52.3/72.4 on MuSiQue/2Wiki).

---

> > ### Author Rebuttal · Reviewer_kN5S · 2026-04-01
> >
> > I thank the authors for a careful and responsive rebuttal. The controlled ablation isolating RICR from evidence format (Q2) is convincing — the drop from 0.523 to 0.342 when the same QA pairs are retrieved via a standard vector store clearly demonstrates that the gains are retrieval-driven, not format-driven. The end-to-end cost accounting (Q4) is also satisfactory.
> >
> > My concerns are partially but not fully resolved on two fronts. First, the contribution remains primarily a well-engineered integration of an inherited representation (GSW) with a chain-following retrieval procedure; the rebuttal clarifies but does not change this. Second, the staged document-arrival experiment (Q3) is appreciated but still falls short of true continual learning — there are no knowledge conflicts, no entity evolution, and no forgetting. I would strongly encourage the authors to narrow the framing accordingly in the final version.
> >
> > On balance, the new evidence tips the paper from clear weaknesses to borderline. I raise my overall score from **3 to 4**.

---

> > > ### Author Response · Authors · 2026-04-02
> > >
> > > Thank you again for the thoughtful follow-up and constructive feedback. We appreciate your note that the rebuttal moved your assessment from 3 to 4, and we will clarify the continual-learning framing in the final version. On our side, the current overall recommendation still appears as 3; OpenReview may require editing the original review separately from the acknowledgement text.

---

### Official Review · Reviewer_KwEj · 2026-03-11

**Soundness:** 3
**Presentation:** 2
**Significance:** 3
**Originality:** 3
**Overall Recommendation:** 4
**Confidence:** 3

**Summary:**

This paper introduces a structured memory for RAG, called Panini. It first constructs a generative semantic workspace (GSW) for the graph-structured memory using documents in corpora. Then it builds two indexes: a sparse index of entities and a dense index of QA pairs. When given a question at inference time, Panini first decomposes the question into subqueries using GPT-4o, and then retrieves entities from the sparse index and QA pairs from the dense index. Then, it retrieves the related substructure from the built GSW. Finally, an LLM (GPT-4o-mini) predicts answers based on the generated chains based on the retrievals.

**Compliance With Llm Reviewing Policy:**

Affirmed.

**Key Questions For Authors:**

Could the authors explain why they used proprietary models for decomposition and answer prediction? Doesn't a smaller model work for these tasks?

Using a proprietary LLM during inference seems to give the proposed method a greater advantage. How do the authors think about this?

**Limitations:**

yes

**Strengths And Weaknesses:**

**Strength**

To address non-parametric continual learning with memory, Panini turns documents into reusable reasoning units (QA experiences) instead of retrieving raw text. This not only reduces context length but also removes noise in long paragraphs. Additionally, it uses graph-structured memory for continual learning in information retrieval, significantly enhancing performance.

In experiments, they use several baselines ranging from sparse & dense retrieval to structure-augmented and agentic RAG. These exhaustive experiments suggest the outperformance and robustness across several benchmarks.

---

**Weakness**

The significant issue is the fairness of computation. When a question is presented, this framework first decomposes it using **GPT-4o**. This is an unfair advantage of advanced LLM compared to competing methods.

Although related work addresses parametric and non-parametric continual learning, it neither compares their approaches to continual setting in information retrieval, such as [1] or [2].

[1] Chen et al., “Continual Learning for Generative Retrieval over Dynamic Corpora”, CIKM 2023.
[2] Ko et al., “When Should Dense Retrievers Be Updated in Evolving Corpora? Detecting Out-of-Distribution Corpora Using GradNormIR”, ACL 2025 Findings.

---

> ### Author Rebuttal · Authors · 2026-03-31
>
> We thank the reviewer for the thoughtful reading, positive assessment, and helpful pointers. We address the main concerns below.
>
> Q1. Why use proprietary models, and does this create an unfair advantage?
>
> We do not believe PANINI’s gains are explained by a proprietary-model advantage. First, in the main experiments all methods use the same GPT-4o-mini answer model, so **PANINI does not benefit from a stronger downstream reader**. We also did not introduce PANINI-specific prompt engineering to create an artificial advantage: the answerer is simply given retrieved evidence and asked to answer from that evidence, rather than being optimized with a special prompt designed to favor PANINI. Second, PANINI’s only additional inference-time LLM call is a single one-shot decomposition step rather than an iterative reasoning loop; this step is small, averaging about 60 tokens per query, and end-to-end query latency on MuSiQue remains 3.3 seconds for PANINI, compared with 4.4 for HippoRAG 2 and 6.72 for Search-R1.
>
> **Yes, smaller/open models do work:** PANINI still outperforms HippoRAG 2 under open-source read-time evaluation with Qwen3-8B as reader and Qwen3-8B+LoRA for decomposition. As reported in Appendix D, PANINI achieves 63.47 average F1 versus 60.02 for HippoRAG 2. On MuSiQue, the gap is 52.43 versus 45.40; on 2Wiki, 70.28 versus 66.93. In other words, PANINI’s advantage is not only preserved but slightly larger in the open-source setting, including relative to the strongest prior baseline, HippoRAG 2. The same appendix also reports open-source Platinum results, where PANINI remains the strongest or near-strongest method under open-source read-time evaluation. We further evaluate fully open-source pipelines in which GSW construction is also done with open models; in Table 10, GPT-OSS-120B construction reaches 48.50 on MuSiQue and 70.65 on 2Wiki. Taken together, these results suggest that the gains come from the structured-memory plus RICR design rather than dependence on proprietary model access.
>
> Q2. How does PANINI relate to continual-learning work in IR?
>
> We appreciate the pointer and agree these papers should be acknowledged. Chen et al. (CIKM 2023) study continual learning for generative retrieval over dynamic corpora, using incremental indexing and memory-augmented training so a parametric retriever can absorb new documents without forgetting older ones. Ko et al. (ACL Findings 2025) study when a dense retriever should be updated as corpora evolve, proposing an unsupervised method to detect out-of-distribution corpora before indexing. These are important related works on updating retrievers under evolving corpora. PANINI instead studies non-parametric continual learning, where the base model is fixed and new knowledge is incorporated by writing new documents into an external structured memory that can be queried immediately.
>
> To make that setting concrete, we also ran a staged document-arrival experiment during the rebuttal period (we provide more detail in our response to reviewer kN5S). We split 100 MuSiQue questions into 10 groups and, at each stage, append the supporting documents for one additional group while keeping distractors present, then evaluate all 100 questions. As expected for a continual learner, overall F1 rises from 22.7 to 82.9 as more evidence becomes available, while PANINI’s answerable-only F1 stays approximately flat at about 83% across stages. The best baseline reaches 66.4 at the final stage, leaving a +16.5 F1 gap. Figure (staged document-arrival experiment): https://anonymous.4open.science/r/panini-rebuttal-2090/plots/controlled_cl_combined.png
>
> We will cite these papers and clarify in the revision that they are complementary to our setting rather than direct baselines.

---

> > ### Author Rebuttal · Reviewer_KwEj · 2026-04-02
> >
> > Thank you for the thoughtful response. The clarification on continual learning in IR is helpful.
> >
> > However, I still have one concern regarding the decomposer. While an open-source decomposer is used, it is trained on MuSiQue and applied deterministically (temperature = 0), so it is unclear how well it generalizes to more diverse or complex queries. Since decomposition directly affects the reasoning chain, this raises questions about whether the observed gains come from the proposed framework or from dataset-specific decomposition.
> >
> > For this reason, I will maintain my original score.

---

> > > ### Author Response · Authors · 2026-04-05
> > >
> > > Thank you for the thoughtful follow-up. We agree that this is an important clarification. In the paper, the open-source Qwen decomposer is trained on a split of MuSiQue train, while evaluation is performed on a completely separate held-out MuSiQue set. Furthermore, we apply this same fixed decomposer unchanged across five benchmarks in Table 9, spanning both single-hop and multi-hop QA with different query styles, where PANINI still outperforms HippoRAG 2 on average across datasets (63.47 vs 60.02). The fact that a single decomposer trained only on MuSiQue transfers effectively to additional benchmarks, covering single-hop, multi-hop, and diverse query styles, indicates that the observed gains are not an artifact of dataset-specific decomposition but reflect the robustness of the framework itself.
> > >
> > > To test your concern more directly, we additionally ran no-finetuning experiments using prompted GPT-OSS decomposers on MuSiQue:
> > >
> > > | Decomposer | Answerer | MuSiQue F1 |
> > > | --- | --- | --- |
> > > | GPT-OSS-120B (prompted, no finetuning) | GPT-OSS-120B | 55.30 |
> > > | GPT-OSS-120B (prompted, no finetuning) | GPT-4o-mini | 52.81 |
> > > | GPT-OSS-20B (prompted, no finetuning) | GPT-OSS-20B | 52.85 |
> > > | GPT-OSS-20B (prompted, no finetuning) | GPT-4o-mini | 52.25 |
> > > | HippoRAG 2 (paper baseline) | GPT-4o-mini | 49.3 |
> > > | PANINI (paper main result) | GPT-4o-mini | 52.3 |
> > >
> > > These prompted open-source decomposers still outperform the strongest baseline, HippoRAG 2, by a clear margin, and are comparable to or better than the main PANINI MuSiQue result. This suggests the gains are again not driven by dataset-specific finetuning, but persist under prompting with open decomposers.

---

### Official Review · Reviewer_SFfq · 2026-03-13

**Soundness:** 3
**Presentation:** 3
**Significance:** 2
**Originality:** 2
**Overall Recommendation:** 4
**Confidence:** 3

**Summary:**

This paper proposes **PANINI**, a non-parametric continual learning framework that stores documents as structured **Generative Semantic Workspaces (GSWs)**. Instead of retrieving document chunks as in traditional retrieval-augmented generation (RAG) systems, PANINI retrieves reasoning chains composed of question–answer pairs stored in the GSW graph. The method introduces a beam-search-based **Reasoning Inference Chain Retrieval (RICR)** algorithm and evaluates it on several single-hop and multi-hop QA benchmarks. The authors argue that structuring knowledge at write time enables more efficient inference-time reasoning and improves answer reliability compared to standard RAG systems.

**Compliance With Llm Reviewing Policy:**

Affirmed.

**Final Justification:**

I remain positive on this paper. The core idea is interesting and technically meaningful, and the rebuttal adequately addressed my main concerns. While some framing and novelty questions remain, I believe the paper is solid and I lean to accept it.

**Key Questions For Authors:**

Please see the weaknesses.

**Limitations:**

yes

**Strengths And Weaknesses:**

**Strengths**
1. Clear motivation: Conventional RAG systems require LLMs to repeatedly process the same document chunks during inference, which may lead to increased computational cost. The paper aims to address this inefficiency by restructuring document knowledge into a more compact representation.
2. Token efficiency improvement: By retrieving compact QA pairs instead of long passages, PANINI reduces the number of tokens passed to the LLM during inference.
3. Generally clear presentation: The paper is generally well written, and the diagrams (e.g., Figure 1 and Figure 2) clearly illustrate the proposed architecture and workflow.

**Weaknesses**
1. Unclear relationship with the original GSW framework
It is unclear whether the process of constructing the **Global Semantic Workspace** (not the individual generative semantic workspace) from continual experiences (Steps 1–3 in Figure 1) is identical to that proposed in **Rajesh et al. (2025)**[1] or whether there are any modifications. This should be clarified explicitly.

In addition, the original GSW work describes the framework as **brain-inspired**, whereas the present paper characterizes PANINI as **human-like**. However, the manuscript does not provide a clear theoretical justification for this claim, which makes the characterization appear somewhat arbitrary.

2. High write-time cost and reliability of LLM-based knowledge extraction
**(a) Scalability and efficiency.**

The computational cost of building GSW structures is likely substantial, but the paper does not quantify the scalability or computational efficiency of this process.

**(b) Reliability of structured knowledge extraction.**

The framework directly uses a general-purpose LLM to extract structured knowledge without any dedicated pretraining or fine-tuning. This raises concerns about the reliability of the generated structures, since LLMs may produce hallucinated or inconsistent outputs. The paper does not provide an evaluation of extraction quality or an analysis of error propagation within the GSW graph.

3. Missing evaluation of retrieval latency
The paper emphasizes **token efficiency**, but does not report **retrieval latency** or **end-to-end inference time**. Since the proposed RICR algorithm relies on beam search over graph structures, and involves multiple retrieval and reranking steps per query, it is unclear whether the method actually reduces inference-time cost in practice. Evaluating retrieval latency and end-to-end QA latency would be necessary to substantiate the efficiency claims.

4. Potential limitation of novelty
The paper would benefit from a clearer explanation of how GSW differs from existing Knowledge Graph (KG) representations or other structured memory architectures. Given the large number of prior studies on KG-based RAG systems for QA [2], the novelty of the proposed approach is not entirely clear from the current presentation.

References
[1] Rajesh S, Holur P, Duan C, et al. Beyond fact retrieval: Episodic memory for rag with generative semantic workspaces. arXiv:2511.07587, 2025.
[2] Ma C, Chen Y, Wu T, et al. Large language models meet knowledge graphs for question answering: Synthesis and opportunities. EMNLP. 2025: 24589-24608.


**Suggestions**
1. Reconsider the framing of “continual learning”
Whether **“continual learning”** is the most appropriate framing for this work is worth further discussion.

In the current framework, the “continual” aspect is limited to the ability to incrementally incorporate new documents into the memory structure. However, this property is also shared by standard vector databases and many existing RAG systems. As such, the term "continual learning" may somewhat overstate the degree of learning involved (even though the paper explicitly mentions the non-parametric setting).

The contribution of this work appears to be closer to **a structured-memory RAG architecture designed for more efficient reasoning**, rather than continual learning in the traditional sense.

Clarifying this positioning would strengthen the conceptual clarity of the paper.

---

> ### Author Rebuttal · Authors · 2026-03-31
>
> We thank the reviewer for the constructive feedback.
>
> Q1. Relationship to prior GSW work, and "human-like" phrasing.
>
> The key point is that PANINI is not a reuse of prior GSW work. Prior GSW work studied episodic memory over continuous narratives, where write-time reconciliation is needed to aggregate state and spatio-temporal changes across events. PANINI instead addresses fact-based multi-hop QA over disparate documents. Here the evidence is atomic: each hop usually needs one specific fact, not aggregation over a long narrative. That task difference changes both what should be written into memory and how it should be read.
>
> PANINI therefore uses per-document memories and avoids exhaustive write-time cross-document reconciliation, because in factual multi-hop QA it creates noisy links that hurt retrieval. Instead, PANINI introduces query-time reconciliation via RICR: the query is decomposed once, and retrieval follows QA chains mechanically through structured memory without further LLM calls.
>
> We agree that the "human-like" phrasing can be softened. Our technical claim is that PANINI incrementally writes new information into structured external memory and reads it by reasoning over that memory, rather than repeatedly reprocessing raw chunks.
>
> Q2. Scalability, write-time cost, and retrieval latency.
>
> Appendix H, reports these numbers on MuSiQue: write cost USD 48.02 with the Batch API versus USD 91.8 for HippoRAG 2; indexing time 100.1 min versus 106.6; end-to-end QA latency 3.3s/query versus 4.4s for HippoRAG 2 and 6.72s for Search-R1. PANINI is faster at read time than the strongest structured and agentic baselines while delivering higher QA accuracy. This is because PANINI makes one small decomposition call (~60 tokens/query) and then performs retrieval over compact QA pairs rather than repeated LLM reasoning at each hop. For a controlled comparison, see our response to reviewer kN5S (Q2).
>
> Q3. Reliability of LLM-based knowledge extraction.
>
> We agree GSW construction is not error-free. Appendix F identifies three recurring failure modes: missing verb-phrase nodes, missing inverse QA links, and imperfect decomposition. To address robustness and reproducibility directly, **Appendix D includes extensive open-source evaluations** across the pipeline. Importantly, PANINI's advantage is not only preserved but slightly larger in the open-source setting: with a Qwen3-8B reader and Qwen3-8B+LoRA decomposition, PANINI reaches 63.47 average F1 versus 60.02 for HippoRAG 2. We also evaluate a fully open-source pipeline by varying only the GSW constructor under the same Qwen3-8B read-time configuration: PANINI improves from 40.24/55.38 (MuSiQue/2Wiki) with Qwen3-8B construction to 48.50/70.65 with GPT-OSS-120B, and the latter still exceeds HippoRAG 2 under the same reader (45.40/66.93). The same pattern appears under missing evidence: on Platinum, PANINI attains substantially stronger refusal accuracy on unanswerable questions than HippoRAG 2 (72.8 vs 58.5). If the extracted structure were systematically inconsistent, it would be difficult to obtain both stronger open-source margins and stronger abstention under missing evidence.
>
> Q4. Difference from knowledge-graph approaches and novelty.
>
> PANINI differs from prior KG-based and structured-memory RAG systems in both representation and retrieval. In many KG-RAG systems, memory is an OpenIE-style graph where entities are nodes and edges are extracted relations between entity pairs. PANINI instead stores document-local concept/event structure with grounded QA pairs, so entities are connected through actions/events and answerable reasoning units rather than only through fixed relation labels. In that sense, it is closer in spirit to frame- or concept-centric resources such as FrameNet [Baker et al., 1998] or ConceptNet [Speer et al., 2017], but generated directly from each document by an LLM rather than from a hand-specified ontology.
>
> This difference matters at retrieval time. In systems such as HippoRAG, the graph mainly serves as a search index: relevance is propagated over generic edges (e.g., with PPR) to surface source passages, and the final LLM still reasons over retrieved chunks. PANINI instead uses the structured representation itself as the retrieval-and-composition substrate. Its concept/event-plus-QA structure supports question-conditioned semantic chaining: after a single decomposition call, RICR follows QA chains by instantiating later sub-questions from earlier hop answers, without further LLM reasoning. PANINI makes only the cross-document connections required by the query. For a controlled comparison over the same QA units, please also see our response to reviewer kN5S (Q2).
>
> On framing, we will clarify that the continual aspect here is incremental non-parametric memory updates without weight changes. Figure 9 already evaluates corpus growth under increasing distractors; for a staged document-arrival experiment, see our response to reviewer kN5S (Q3).

---

> > ### Author Rebuttal · Reviewer_SFfq · 2026-04-01
> >
> > Thanks the authors for the clear and thorough rebuttal.
> >
> > The rebuttal helps clarify the distinction between PANINI and prior GSW work,  the positioning of the continual-learning framing, and provides useful evidence on write-time cost and end-to-end latency. I also appreciate the added discussion of extraction failure modes and the representation/retrieval differences from KG-based approaches.
> > I will maintain my current positive score.

---

### Decision · Program_Chairs · 2026-04-30

**Decision:**

Accept (regular)

**Comment:**

This work proposes a human-like non-parametric continual learning framework that stores documents as structured Generative Semantic Workspaces (GSWs), enabling an LLM to reconstruct experienced situations and mine latent knowledge through reasoning-grounded inference chains over the network. The reviewers acknowledge the paper’s motivation, empirical results, and discussion of extraction failure modes. During the rebuttal, the authors addressed most concerns regarding the GSW design and the continual learning setting. Overall, this is a good work.